# FreqCa: Accelerating Diffusion Models via Frequency-Aware Caching

## Abstract

The application of diffusion transformers is suffering from their significant inference costs. Recently, feature caching has been proposed to solve this problem by reusing features from previous timesteps, thereby skipping computation in future timesteps. However, previous feature caching assumes that features in adjacent timesteps are similar or continuous, which does not always hold in all settings. To investigate this, this paper begins with an analysis from the frequency domain, which reveal that *different frequency bands in the features of diffusion models exhibit different dynamics across timesteps.* Concretely, low-frequency components, which decide the structure of images, exhibit higher *similarity* but poor continuity. In contrast, the high-frequency bands, which decode the details of images, show significant continuity but poor similarity. These interesting observations motivate us to propose **Freq**uency-aware **Ca**ching (*FreqCa*) which directly reuses features of low-frequency components based on their similarity, while using a second-order Hermite interpolator to predict the volatile high-frequency ones based on its continuity. Besides, we further propose to cache Cumulative Residual Feature (CRF) instead of the features in all the layers, which reduces the memory footprint of feature caching by **99%**. Extensive experiments on FLUX.1-dev, FLUX.1-Kontext-dev, Qwen-Image, and Qwen-Image-Edit demonstrate its effectiveness in both generation and editing. *Codes are available in the supplementary materials and will be released on GitHub.*

## 1 Introduction

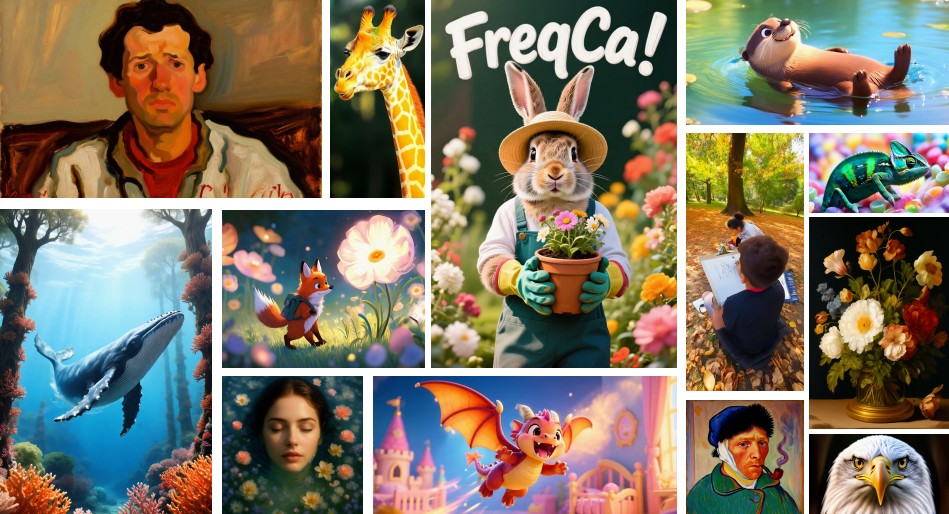

Figure 1: Images sampled by Qwen-image with *FreqCa* with 7.14× acceleration.

Diffusion Models (DMs) have achieved remarkable success in generative tasks like image synthesis and video generation (Ho et al., 2020a; Rombach et al., 2022; Blattmann et al., 2023). The recent introduction of Diffusion Transformers (Peebles & Xie, 2023a) has further advanced generation

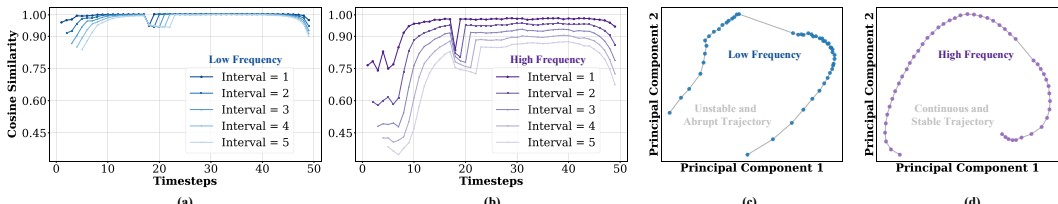

Figure 2: **Analysis from the frequency perspective. (a)-(b)**:Temporal similarity analysis using cosine similarity for low-frequency and high-frequency components across different step intervals. **(c)-(d)**: Feature trajectory visualized via Principal Component Analysis (PCA).

quality and diversity, establishing them as the predominant architecture for large-scale visual content creation. However, diffusion transformers typically rely on a stack of heavy transformer blocks and multi-step sampling, making computational efficiency a critical bottleneck for their practical deployment. To address this, the paradigm of feature caching has emerged, which exploits the high temporal redundancy between adjacent timesteps for acceleration (Ma et al., 2024; Li et al., 2023a; Selvaraju et al., 2024; Chen et al., 2024; Zou et al., 2025; 2024).

**The debate of caching paradigms.** Feature caching has gradually emerged into two different paradigms. The paradigm of "Cache-Then-Reuse" assumes that the features of DM in adjacent timesteps are highly **similar** and thus proposes to directly **reuse** the features of previous timesteps in the future timesteps (Selvaraju et al., 2024). In contrast, the paradigm of "Cache-Then-Forecast" assumes that features of DM are "continuous" and thus proposes to forecast features in the future timesteps based on features in the previous timesteps with non-parametric predictors such as Taylor expansion. Although the paradigm "Cache-Then-Forecast" tends to show better performance in recent works, their assumption of continuity does not always hold perfectly. For instance, Liu *et al.* demonstrates that features of FLUX are not high-order continuous, making TaylorSeer degenerate into a linear prediction method and thus suffer from quality loss (Liu et al., 2025a). Based on these findings, this paper begins with an in-depth analysis of the temporal dynamics of diffusion models.

**Analysis from the frequency perspective.** In classical image processing, the high-frequency and low-frequency components of images are usually considered as carrying different semantic information, which motivates us to study the dynamics of high-frequency and low-frequency components in the feature of diffusion models separately. As shown in Figure 2(a)-(b), surprisingly, we find different frequency exhibits significantly different dynamics. Concretely, the similarity of low-frequency is higher than *0.90* at most timesteps, while the high-frequency exhibits clearly low similarity. On the other hand, as shown in Figure 2(c)-(d), the feature trajectory of high-frequency shows perfect stability and continuity, while the feature trajectory of low-frequency is unstable and accompanied by sudden mutation, indicating that the high-frequency information can be accurately predicted while the low-frequency information fails.

Based on the above observations, this paper introduces **Freq**uency-aware Feature **Ca**ching (**FreqCa**), which aims to decouple the frequency of the features in diffusion models and treat them in different paradigms. Concretely, *FreqCa* applies any frequency decomposition (*e.g, Fourier Transformation*) to the cached features. For the low-frequency bands, we directly reuse them in the future timesteps because of their *high similarity*. For the high-frequency bands, we predict their values in the future timesteps with any sequential predictor (*Hermite polynomial predictor*) for their *good continuity*. Then, in the future timesteps, *FreqCa* reconstructs the features based on the reused low-frequency bands and predicted high-frequency bands, enabling it to skip the computation over diffusion transformers, achieving the best cooperation between the previous caching paradigms.

**Memory-Efficient Feature Caching.** The previous caching methods usually cache all the features from attention and FFN layers, leading to significant memory costs (*e.g,* $\geq$ 10G memory costs on FLUX in ToCa), preventing feature caching methods from real-world applications. As discussed by Veit et al. (2016), neural networks with residual connections can be considered as an ensemble of features in all blocks, which motivates us to propose caching the Cumulative Residual Feature (CRF), *i.e.,* the cumulative features of all the residual connections from attention and FFN blocks. This trick helps us reduce the memory footprint from caching $2 \times L$ ($L$ indicates layer counts) features into a single feature vector, slashing cache memory usage by up to 99%. Besides, it also reduces the number of frequency (reverse) decomposition operations by $2L$ times, making them account for only $\leq 0.01\%$ latency costs during the whole diffusion process.

In summary, this contribution of this paper is as follows.

- **Frequency-Aware Feature Caching:** Motivated by the difference in similarity and continuity of different frequency bands, we propose *FreqCa*, which applies different feature caching methods to different frequencies, unifying the two previous caching paradigms.

- **Memory-Efficient Feature Caching** By caching only the Cumulative Residual Feature(CRF), *FreqCa* achieves $\mathcal{O}(1)$ memory complexity, slashing Cache memory usage to a mere **1%** of prior approaches without fidelity loss, enabling high-quality acceleration on consumer hardware.

- **State-of-the-art generalization and performance:** Across text-to-image generation and image editing tasks, *FreqCa* consistently delivers 6–7$\times$ acceleration with quality degradation below 2%, outperforming existing methods and demonstrating strong robustness and practicality.

## 2 METHOD

### 2.1 PRELIMINARY

#### 2.1.1 DIFFUSION TRANSFORMER ARCHITECTURE.

The Diffusion Transformer (DiT) (Peebles & Xie, 2023a) employs a hierarchical structure $\mathcal{G} = g_1 \circ g_2 \circ \cdots \circ g_L$, where each module $g_l = \mathcal{F}_{\text{SA}}^l \circ \mathcal{F}_{\text{CA}}^l \circ \mathcal{F}_{\text{MLP}}^l$ is composed of self-attention (SA), cross-attention (CA), and multilayer perceptron (MLP) components. In DiT, these components are dynamically adapted over time to handle different noise levels during the image generation process. The input $\mathbf{x}_t = \{x_i\}_{i=1}^{H \times W}$ is represented as a sequence of tokens corresponding to image patches. Each module integrates information through residual connections of the form $\mathcal{F}(\mathbf{x}) = \mathbf{x} + \text{AdaLN} \circ f(\mathbf{x})$, where AdaLN denotes adaptive layer normalization, which stabilizes training and improves learning effectiveness.

#### 2.1.2 FEATURE CACHING IN DIFFUSION MODELS

Feature caching is designed to speed up diffusion inference by reducing repeated computations across nearby timesteps. Traditional caching follows a simple *reuse* strategy, where the feature map obtained at timestep $t$ is stored and directly applied to earlier latent states without recomputation:

$$\mathcal{F}(x_{t-k}^l) \approx \mathcal{F}(x_t^l), \quad k \in \{1, \dots, N-1\}. \tag{1}$$

Although this scheme provides an idealized $(N-1)$-fold acceleration, it assumes that features remain static across time, which is not true for diffusion processes. As a result, the approximation error grows quickly as $N$ increases. To address this issue, recent approaches adopt a *forecasting-based* caching mechanism. Instead of treating features as invariant, they interpret the caching problem as predicting a smooth temporal trajectory of features. Using finite-difference estimates or low-order polynomial trends, the feature at timestep $(t-k)$ can be estimated via a truncated Taylor-like series:

$$\mathcal{F}_{\text{pred},m}(x_{t-k}^l) = \mathcal{F}(x_t^l) + \sum_{i=1}^{m} \frac{\Delta^i \mathcal{F}(x_t^l)}{i! \, N^i}(-k)^i, \tag{2}$$

where $\Delta^i \mathcal{F}(x_t^l)$ represents the $i$-th order discrete difference. This perspective shifts caching from direct copying to temporal prediction, enabling more accurate feature estimates across timesteps.

#### 2.1.3 FREQUENCY DECOMPOSITION METHODS

Frequency decomposition, through methods like the Fast Fourier Transform (FFT) and Discrete Cosine Transform (DCT), is a powerful technique for decoupling signals into distinct components. This process separates a signal into its **low-frequency** components, which typically represent global structure and smooth layouts, and its **high-frequency** components, which correspond to fine-grained details and sharp edges. In the context of diffusion models, this decoupling allows us to differentiate between stable, foundational structures and volatile, transient details along the generative trajectory.

### 2.2 FREQUENCY-AWARE CACHE ACCELERATION FRAMEWORK

In this section, we introduce the **FreqCa (Frequency-aware Feature Caching)** framework, which is built upon three key components: (i) performing frequency decomposition on the feature to be

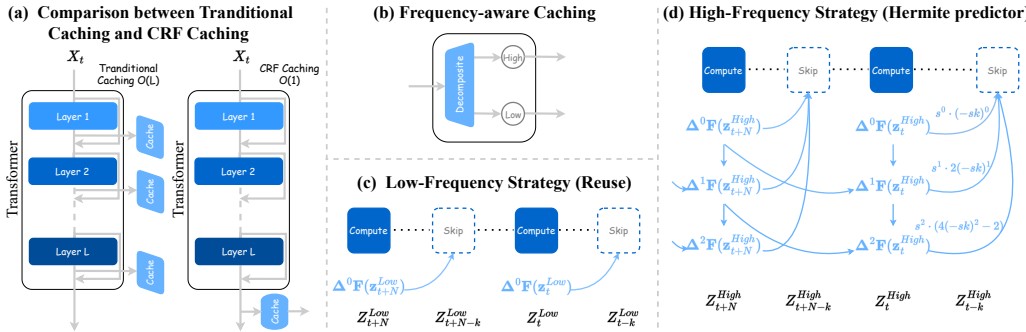

Figure 3: **Overview of the FreqCa framework.** *(a) CRF Caching* : Instead of caching features at every layer, we cache only the single Cumulative Residual Feature (CRF) at the end. *(b) Frequency-aware Caching*: The cached features are separated into low- and high-frequency bands using frequency decomposition techniques such as FFT or DCT. *(c) Low-Frequency Strategy*: Low-frequency component is directly reused from the prior step. *(d) High-Frequency Strategy*: High-frequency component is forecasted using a Hermite predictor fitted on the last two activated steps.

cached and applying separate strategies for its low- and high-frequency components; (ii) employing a nonlinear Hermite-polynomial-based predictor for the high-frequency part to improve prediction accuracy; and (iii) identifying the Cumulative Residual Feature (CRF) as a novel, highly efficient single-tensor caching target that encapsulates the entire transformation history of the model. *Please refer to the Appendix for additional analysis and results.*

**1. Frequency-Decomposed Caching and Prediction Strategy.** Our differentiated caching strategy is motivated by the distinct temporal dynamics of frequency components. Low-frequency components exhibit high similarity but low continuity, making them stable but difficult to predict. Conversely, high-frequency components are less similar but more continuous, making them volatile yet predictable along a trajectory. This key difference means that a one-size-fits-all approach is not the best and that a differentiated strategy is required.

To implement this, we decompose the feature $\mathbf{z}_t$ (defined as the CRF) into its constituent parts. We utilize a frequency transform $\mathcal{T}$ (e.g., DCT) and a radial mask $\mathbf{M}$ with a cutoff frequency $\tau$ to separate the spectrum:

$$\mathbf{z}_t^{\text{low}} = \mathcal{T}^{-1}(\mathcal{T}(\mathbf{z}_t) \odot \mathbf{M}_{\text{low}}), \quad \mathbf{z}_t^{\text{high}} = \mathcal{T}^{-1}(\mathcal{T}(\mathbf{z}_t) \odot \mathbf{M}_{\text{high}}).$$

Based on their dynamics, we apply tailored strategies. Given the structural stability of the low-frequency component, we apply a direct reuse strategy from the most recent activation step $t_{prev}$:

$$\widehat{\mathbf{z}}_t^{\text{low}} = \mathbf{z}_{t_{prev}}^{\text{low}}.$$

For the volatile high-frequency component, simple reuse leads to detail degradation. Instead, we model its trajectory using a second-order Hermite interpolator. We employ an efficient **finite-difference** approach. Let $\Delta\mathbf{F}^{(1)}$ and $\Delta\mathbf{F}^{(2)}$ represent the first- and second-order discrete differences computed from the high-frequency features of previous full-computation steps. The feature at the current prediction step is estimated as:

$$\widehat{\mathbf{z}}_t^{\text{high}}(d) = \mathbf{z}_{t_{prev}}^{\text{high}} + \alpha_1(d)\Delta\mathbf{F}^{(1)} + \alpha_2(d)\Delta\mathbf{F}^{(2)},$$

where $d$ is the interval distance from the previous step, and $\alpha_k(d)$ are coefficients derived from Hermite polynomials. This closed-form predictor allows for accurate high-frequency reconstruction with negligible computational overhead. The final predicted feature is obtained by recombining the components: $\widehat{\mathbf{z}}_t = \widehat{\mathbf{z}}_t^{\text{low}} + \widehat{\mathbf{z}}_t^{\text{high}}$.

**2. Cumulative Residual Feature (CRF)** At its core, a Diffusion Transformer (DiT) is a deep stack of $L$ residual blocks. The transformation at each block $l$ is not a replacement but an incremental update, as described by the standard residual connection: $\mathbf{h}^{(l+1)} = \mathbf{h}^{(l)} + \mathcal{F}^{(l)}(\mathbf{h}^{(l)}, t)$, where $\mathcal{F}^{(l)}(\cdot, t)$ denotes the transformation module at layer $l$ (including Attention, and MLP), which is dynamically modulated by the diffusion timestep $t$ (e.g., through AdaLN).

The structure of the DiT's final output is thus revealed: $\phi_L(\mathbf{x}_t) = \mathbf{h}^{(0)} + \sum_{l=0}^{L-1} \mathcal{F}^{(l)}(\mathbf{h}^{(l)}, t)$. This formulation shows that the final output is not just another intermediate feature, but the **accumulation of the initial input and all subsequent residual updates**. We define this special output $\mathbf{z}_t \triangleq \phi_L(\mathbf{x}_t)$ and name it the **Cumulative Residual Feature (CRF)**, reflecting its composite nature.

This insight leads to a more memory-efficient strategy. While conventional layer-wise caching must store all intermediate features $\{\mathbf{h}^{(l)}\}_{l=0}^{L-1}$, our approach uses the fact that the CRF already contains the entire transformation history. We use this single, globally fused tensor as a highly efficient replacement for the full feature set. As shown in Figure 4, caching only the CRF achieves nearly identical reconstruction fidelity to full layer-wise caching, incurring only a 4% higher MSE on average, which confirms that the CRF acts as a near-lossless compression of the entire computational path. This makes it an ideal lightweight caching target, enabling a revolutionary reduction in memory complexity from $\mathcal{O}(L)$ to $\mathcal{O}(1)$ without a meaningful sacrifice in quality.

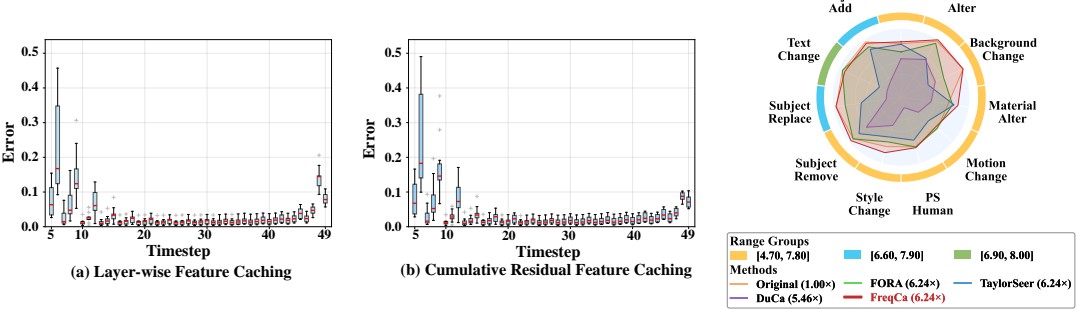

Figure 4: Box plots of Mean Squared Error (MSE) between ground-truth and predicted features per timestep. (a) layer-wise feature caching and (b) cumulative residual feature (CRF) caching.

Figure 5: Gedit Benchmark on Qwen-Image-Edit, *FreqCa* outperforms most baselines.

## 3 Experiment

### 3.1 Experiment Settings

**Model Configurations.** The experiments are conducted on four state-of-the-art visual generative models—**FLUX.1-dev** (Labs, 2024), **Qwen-Image** (Liu et al., 2023a), **FLUX.1-Kontext-dev** (Zhang & Agrawala, 2025), and **Qwen-Image-Edit** (Salimans & Ho, 2022).

**Evaluation and Metrics.** For the text-to-image generation evaluation, we adopt the DrawBench (Saharia et al., 2022) benchmark. The generated samples are systematically evaluated using ImageReward (Xu et al., 2023) and CLIP Score (Hessel et al., 2021), which jointly measure image quality and text–image semantic alignment. To assess visual fidelity, we further employ PSNR, SSIM (Wang et al., 2004) and LPIPS (Zhang et al., 2018), thereby capturing both pixel-level similarity and perceptual consistency. Additionally, we evaluate general-purpose image editing using the GEdit benchmark (Wang et al., 2024), which systematically assesses instruction-driven editing fidelity and alignment to target modifications under textual and visual guidance.

### 3.2 Text-to-Image Generation

#### 3.2.1 FLUX.1-dev

On FLUX.1-dev, FreqCa consistently outperforms state-of-the-art acceleration methods across different speedup levels. At 2.63× speedup, FreqCa achieves an ImageReward of 1.00, clearly out-

performing FORA and TeaCache. At 4.99× speedup, it maintains lossless quality. Even under 6.24× speedup, FreqCa achieves only a 2% drop in ImageReward (0.97), while TaylorSeer suffers a degradation of 13.1%. FreqCa also achieves 2.00× speedup on distilled FLUX.1-schnell while improving ImageReward from 0.93 to 0.95.

Table 1: **Quantitative comparison in text-to-image generation** for FLUX.1-dev and FLUX.1-schnell(a distilled version). Best results are highlighted in **bold**, and second-best are underlined.

| Method | Acceleration | | | | Quality Metrics | | Perceptual Metrics | | |
|---|---|---|---|---|---|---|---|---|---|
| | Latency(s) ↓ | Speed ↑ | FLOPs(T) ↓ | Speed ↑ | ImageReward↑ | CLIP↑ | PSNR↑ | SSIM↑ | LPIPS↓ |
| **[dev]: 50 steps** | 23.24 (+0.0%) | 1.00× | 3726.87 | 1.00× | 0.99 (+0.0%) | 32.64 (+0.0%) | ∞ | 1.00 | 0.00 |
| 60% **steps** | 14.12 (-39.2%) | 1.65× | 2236.12 | 1.67× | 0.97 (-2.0%) | 32.66 (+0.1%) | 30.31 | 0.78 | 0.25 |
| 50% **steps** | 11.82 (-49.1%) | 1.97× | 1863.44 | 2.00× | 0.97 (-2.0%) | 32.57 (-0.2%) | 29.56 | 0.73 | 0.31 |
| **PAB** | 17.84 (-23.2%) | 1.30× | 3013.13 | 1.24× | 0.95 (-4.0%) | 32.55 (-0.3%) | 28.84 | 0.67 | 0.40 |
| **DBCache** | 16.88 (-27.4%) | 1.38× | 2384.29 | 1.56× | 1.01 (+2.0%) | 32.53 (-0.3%) | 33.86 | 0.87 | 0.12 |
| **FORA** ($\mathcal{N}$=3) | **9.06** (-61.0%) | **2.57×** | 1267.89 | **2.94×** | 0.93 (-6.1%) | **32.89** (+0.8%) | 28.86 | 0.66 | 0.40 |
| **TeaCache** (*l*=0.6) | 9.13 (-60.7%) | 2.55× | 1342.20 | 2.78× | 0.91 (-8.1%) | 32.11 (-1.6%) | 29.03 | 0.68 | 0.40 |
| **TaylorSeer** ($\mathcal{N}$=3, $O$=2) | 10.16 (-56.3%) | 2.29× | 1416.92 | 2.63× | **1.01** (+2.0%) | 32.86 (+0.7%) | 30.77 | 0.78 | 0.23 |
| **FreqCa** ($\mathcal{N}$=3) | 9.37 (-59.7%) | 2.48× | 1417.40 | 2.63× | 1.00 (+1.0%) | 32.61 (-0.1%) | **33.03** | **0.86** | **0.13** |
| **FORA**($\mathcal{N}$=5) † | 5.97 (-74.3%) | 3.89× | 820.80 | 4.54× | 0.82 (-17.2%) | 32.48 (-0.5%) | 28.44 | 0.60 | 0.50 |
| **ToCa**($\mathcal{N}$=8, $\mathcal{R}$=75%) | 12.39 (-46.7%) | 1.88× | 829.86 | 4.49× | 0.95 (-4.0%) | 32.60 (-0.1%) | 29.07 | 0.64 | 0.43 |
| **DuCa**($\mathcal{N}$=8, $\mathcal{R}$=70%) | 9.40 (-59.6%) | 2.47× | 858.27 | 4.34× | 0.94 (-5.1%) | 32.58 (-0.2%) | 29.06 | 0.64 | 0.43 |
| **TeaCache**(*l*=1.0) † | 7.07 (-69.6%) | 3.29× | 820.55 | 4.54× | 0.84 (-15.2%) | 31.88 (-2.3%) | 28.61 | 0.64 | 0.48 |
| **TaylorSeer**($\mathcal{N}$=6, $O$=2) | 6.73 (-71.0%) | 3.45× | 746.28 | **4.99×** | **1.00** (+1.0%) | 32.91 (+0.8%) | 28.94 | 0.66 | 0.40 |
| **FreqCa** ($\mathcal{N}$=7) | **5.19** (-77.7%) | **4.48×** | **746.03** | **4.99×** | **1.01** (+2.0%) | **32.98** (+1.0%) | 30.00 | **0.71** | **0.31** |
| **FORA**($\mathcal{N}$=7) † | 5.09 (-78.1%) | 4.57× | **597.25** | **6.24×** | 0.68 (-31.3%) | 31.90 (-2.3%) | 28.32 | 0.59 | 0.54 |
| **ToCa**($\mathcal{N}$=12, $\mathcal{R}$=85%) † | 9.82 (-57.7%) | 2.37× | 618.57 | 6.02× | 0.80 (-19.2%) | 32.32 (-1.0%) | 28.70 | 0.60 | 0.52 |
| **DuCa**($\mathcal{N}$=12, $\mathcal{R}$=80%) † | 7.74 (-66.7%) | 3.00× | 646.97 | 5.76× | 0.77 (-22.2%) | 32.20 (-1.3%) | 28.71 | 0.60 | 0.53 |
| **TeaCache**(*l*=1.4) † | 6.14 (-73.6%) | 3.79× | 671.51 | 5.55× | 0.74 (-25.3%) | 31.78 (-2.6%) | 28.12 | 0.48 | 0.68 |
| **TaylorSeer**($\mathcal{N}$=9, $O$=2) † | 5.85 (-74.8%) | 3.97× | **597.25** | **6.24×** | 0.86 (-13.1%) | 32.04 (-1.8%) | 28.38 | 0.59 | 0.51 |
| **FreqCa** ($\mathcal{N}$=10) | **4.25** (-81.7%) | **5.47×** | 597.47 | **6.24×** | **0.97** (-2.0%) | **32.53** (-0.3%) | 28.77 | 0.62 | **0.43** |
| **[schnell]: 4 steps** | 2.17 (+0.0%) | 1.00× | 278.41 | 1.00× | 0.93 (+0.0%) | 34.09 (+0.0%) | ∞ | 1.00 | 0.00 |
| **FreqCa** ($\mathcal{N}$=3): **4 steps** | 1.29 (-40.6%) | 1.68× | 139.23 | 2.00× | 0.95 (+2.2%) | 34.47 (+1.1%) | 30.30 | 0.81 | 0.16 |

- † Methods exhibit significant degradation in image quality.
- Gray: Baseline-relative degradation in quality and gains in latency. Blue: **FreqCa** achieves minimal degradation with large latency gains.

### 3.2.2 QWEN-IMAGE

On Qwen-Image, FreqCa demonstrates superior performance across different acceleration levels. At 5.00× speedup, FreqCa achieves an ImageReward of 1.20, outperforming TaylorSeer (1.01). At 7.14× speedup, FreqCa shows only a 18.4% drop in ImageReward (1.02), while TaylorSeer suffers a 41.6% quality loss (0.73). FreqCa achieves 4.00× speedup on distilled Qwen-Image-Lightning with minimal quality degradation. A visual comparison of these results is provided in Fig. 10.

Table 2: **Quantitative comparison in text-to-image generation** for Qwen-Image and Qwen-Image-Lightning (a distilled version). Best results are highlighted in **bold**, and second-best are underlined.

| Method | Acceleration | | | | Quality Metrics | | Perceptual Metrics | | |
|---|---|---|---|---|---|---|---|---|---|
| | Latency(s) ↓ | Speed ↑ | FLOPs(T) ↓ | Speed ↑ | ImageReward↑ | CLIP↑ | PSNR↑ | SSIM↑ | LPIPS↓ |
| 50 **steps** | 127.40 (+0.0%) | 1.00× | 12917.56 | 1.00× | 1.25 (+0.0%) | 35.59 (+0.0%) | ∞ | 1.00 | 0.00 |
| 50% **steps** | 64.10 (-49.6%) | 1.99× | 6458.78 | 2.00× | 1.20 (-4.0%) | 35.31 (-0.6%) | 30.54 | 0.75 | 0.28 |
| 20% **steps** † | 25.92 (-79.6%) | 4.89× | 2583.51 | 5.00× | 0.94 (-26.4%) | 34.95 (-1.0%) | 28.59 | 0.61 | 0.52 |
| **FORA**($\mathcal{N}$=4)† | 38.43 (-70.0%) | 3.32× | 3359.99 | 3.84× | 0.93 (-25.6%) | 34.40 (-3.3%) | 28.66 | 0.59 | 0.51 |
| **ToCa**($\mathcal{N}$=8, $\mathcal{R}$=75%)† | 61.37 (-51.8%) | 2.08× | 2991.34 | 4.32× | 1.02 (-18.4%) | 34.96 (-1.8%) | 28.93 | 0.63 | 0.44 |
| **DuCa**($\mathcal{N}$=9, $\mathcal{R}$=80%)† | 34.73 (-72.7%) | 3.67× | 2958.13 | 4.37× | 0.77 (-38.4%) | 34.62 (-2.7%) | 28.45 | 0.58 | 0.55 |
| **TaylorSeer**($\mathcal{N}$=6) | 30.75 (-75.9%) | 4.14× | **2583.97** | **5.00×** | 1.01 (-19.2%) | 34.71 (-2.5%) | 28.58 | 0.62 | 0.46 |
| **FreqCa**($\mathcal{N}$=6) | **29.80** (-76.6%) | **4.28×** | 2584.70 | **5.00×** | **1.20** (-4.0%) | **35.39** (-0.6%) | **29.67** | **0.78** | **0.33** |
| **FORA**($\mathcal{N}$=6)† | 28.69 (-77.5%) | 4.44× | 2326.74 | 5.55× | 0.48 (-61.6%) | 33.34 (-6.3%) | 28.48 | 0.55 | 0.59 |
| **ToCa**($\mathcal{N}$=12, $\mathcal{R}$=85%)† | 50.95 (-60.0%) | 2.50× | 2406.20 | 5.37× | 0.55 (-56.0%) | 34.08 (-4.2%) | 28.69 | 0.57 | 0.53 |
| **DuCa**($\mathcal{N}$=12, $\mathcal{R}$=90%)† | 28.57 (-77.6%) | 4.46× | 2171.56 | 5.95× | 0.41 (-67.2%) | 33.38 (-6.2%) | 28.38 | 0.57 | 0.60 |
| **TaylorSeer**($\mathcal{N}$=9)† | 24.64 (-80.7%) | 5.17× | 2067.29 | 6.25× | 0.73 (-41.6%) | 32.97 (-7.4%) | 28.25 | 0.56 | 0.58 |
| **FreqCa**($\mathcal{N}$=10) | **22.45** (-82.4%) | **5.68×** | **1809.38** | **7.14×** | **1.02** (-18.4%) | **35.00** (-1.7%) | **28.86** | **0.64** | **0.44** |
| **Qwen-Image-Lightning: 8 steps** | 7.27 (+0.0%) | 1.00× | 560.96 | 1.00× | 1.30 (+0.0%) | 35.26 (+0.0%) | ∞ | 1.00 | 0.00 |
| **FreqCa** ($\mathcal{N}$=2): **8 steps** | 5.29 (-27.2%) | 1.37× | 350.61 | 1.60× | 1.29 (-0.8%) | 35.23 (-0.1%) | 33.01 | 0.83 | 0.12 |
| **FreqCa** ($\mathcal{N}$=3): **8 steps** | 4.63 (-36.3%) | 1.57× | 282.26 | 1.99× | 1.28 (-1.5%) | 35.10 (-0.5%) | 31.36 | 0.77 | 0.18 |
| **FreqCa** ($\mathcal{N}$=4): **8 steps** | 3.29 (-54.9%) | 2.21× | 140.25 | 4.00× | 1.29 (-0.8%) | 35.94 (+1.9%) | 29.25 | 0.62 | 0.36 |

- † Methods exhibit significant degradation in image quality.
- Gray: Baseline-relative degradation in quality and gains in latency. Blue: **FreqCa** achieves minimal degradation with large latency gains.

### 3.3 IMAGE EDITING

#### 3.3.1 FLUX.1-KONTEXT-DEV

On FLUX.1-Kontext-dev, FreqCa outperforms other acceleration methods. At 5.00× speedup, FreqCa achieves a Q_O score of 6.195, outperforming ToCa (6.125). At 6.24× speedup, FreqCa shows only a 0.4% drop in Q_O score, demonstrating better perceptual fidelity.

Table 3: **Quantitative comparison of text-to-image generation** for FLUX.1-Kontext-dev. Best results are highlighted in **bold**, and second-best results are underlined.

| Method | Acceleration | | | | GEdit-EN (Full) | | |
|---|---|---|---|---|---|---|---|
| | Latency(s) ↓ | Speed ↑ | FLOPs(T) ↓ | Speed ↑ | Q_SC ↑ | Q_PQ ↑ | Q_O ↑ |
| **[Kontext]50 steps** | 50.20 (+0.0%) | 1.00× | 8299.54 | 1.00× | 6.481 | 7.331 | 6.213 (+0.0%) |
| **50% steps** † | 25.42 (-49.4%) | 1.97× | 4149.77 | 2.00× | 6.544 | 7.286 | 6.253 (+0.6%) |
| **20% steps** † | 10.47 (-79.1%) | 4.79× | 1659.91 | 5.00× | 6.603 | 7.184 | 6.286 (+1.2%) |
| **ToCa**($\mathcal{N}$=8, $\mathcal{R}$=70%) | 29.56 (-41.1%) | 1.70× | 1841.35 | 4.51× | 6.432 | 7.256 | 6.125 (-1.4%) |
| **DuCa**($\mathcal{N}$=8, $\mathcal{R}$=60%) | 13.12 (-73.9%) | 3.83× | 1669.08 | 4.97× | 6.469 | 7.195 | 6.150 (-1.0%) |
| **TaylorSeer**($\mathcal{N}$=6, $\mathcal{O}$=2) | 13.95 (-72.2%) | 3.60× | **1660.95** | 5.00× | 6.477 | **7.296** | 6.170 (-0.7%) |
| **FreqCa**($\mathcal{N}$=7) | **10.77** (-78.5%) | **4.66×** | 1661.37 | 5.00× | **6.480** | 7.271 | **6.195** (-0.3%) |
| **ToCa**($\mathcal{N}$=12, $\mathcal{R}$=75%) | 20.72 (-58.7%) | 2.42× | 1359.61 | 6.10× | 6.393 | 6.919 | 6.041 (-2.8%) |
| **DuCa**($\mathcal{N}$=12, $\mathcal{R}$=70%) | 10.39 (-79.3%) | 4.83× | 1376.55 | 6.03× | 6.597 | 7.005 | 6.173 (-0.6%) |
| **TaylorSeer**($\mathcal{N}$=9, $\mathcal{O}$=2) | 12.05 (-76.0%) | 4.17× | **1329.02** | 6.24× | 6.407 | 6.995 | 6.074 (-2.2%) |
| **FreqCa**($\mathcal{N}$=10) | **8.74** (-82.6%) | **5.74×** | 1329.33 | 6.24× | **6.550** | **7.160** | **6.190** (-0.4%) |

- † Methods exhibit significant degradation in image quality; **Q_SC**: semantic consistency, **Q_PQ**: perceptual quality, **Q_O**: overall score.
- Gray: Baseline-relative degradation in quality and gains in latency. Blue: **FreqCa** achieves minimal degradation with large latency gains.

#### 3.3.2 QWEN-IMAGE-EDIT

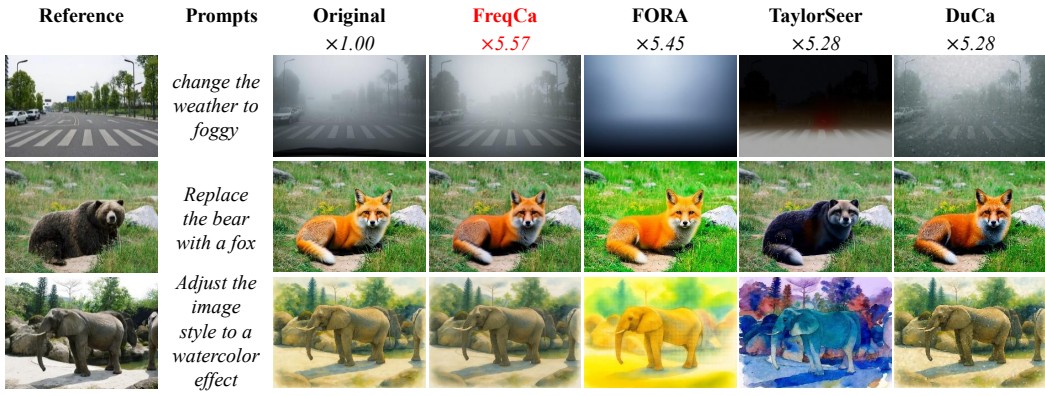

Figure 6: On Qwen-Image-Edit, *FreqCa* delivers higher speedup with near-original editing quality

Table 4: **Quantitative comparison of image editing** using Qwen-Image-Edit. Best results are highlighted in **bold**, and second-best results are underlined.

| Method | Acceleration | | | | GEdit-CN (Full) | | | GEdit-EN (Full) | | |
|---|---|---|---|---|---|---|---|---|---|---|
| | Latency(s) ↓ | Speed ↑ | FLOPs(T) ↓ | Speed ↑ | Q_SC ↑ | Q_PQ ↑ | Q_O ↑ | Q_SC ↑ | Q_PQ ↑ | Q_O ↑ |
| **[full]50 steps** | 284.51 (+0.0%) | 1.00× | 28190.88 | 1.00× | 7.68 | 7.51 | 7.41 (+0.0%) | 7.82 | 7.54 | 7.54 (+0.0%) |
| **50% steps** † | 143.29 (-49.6%) | 1.99× | 14095.44 | 2.00× | 7.70 | 7.53 | 7.44 (+0.4%) | 7.77 | 7.52 | 7.47 (-0.9%) |
| **20% steps** † | 58.45 (-79.5%) | 4.87× | 5638.18 | 5.00× | 7.65 | 7.42 | 7.35 (-0.8%) | 7.73 | 7.46 | 7.44 (-1.3%) |
| **FORA**($\mathcal{N}$=5) | 63.15 (-77.8%) | 4.51× | 5643.13 | 5.00× | 7.60 | 7.31 | 7.25 (-2.2%) | 7.62 | 7.34 | 7.28 (-3.4%) |
| **DuCa**($\mathcal{N}$=7, $\mathcal{R}$=95%) | 69.54 (-75.5%) | 4.09× | 5699.89 | 4.95× | 7.73 | 7.44 | 7.44 (+0.4%) | 7.80 | 7.40 | 7.45 (-1.2%) |
| **TaylorSeer**($\mathcal{N}$=6, $\mathcal{O}$=2) | 65.66 (-76.9%) | 4.33× | 5643.13 | 5.00× | 7.25 | 7.09 | 6.92 (-6.6%) | 7.26 | 7.14 | 6.89 (-8.6%) |
| **FreqCa**($\mathcal{N}$=6) | 62.89 (-77.9%) | 4.52× | 5642.24 | 5.00× | **7.75** | **7.54** | **7.49** (+1.1%) | **7.83** | **7.48** | **7.52** (-0.3%) |
| **FORA**($\mathcal{N}$=7) | 52.20 (-81.7%) | 5.45× | 4515.74 | 6.24× | 7.42 | 7.13 | 7.06 (-4.7%) | 7.43 | **7.19** | 7.06 (-6.3%) |
| **DuCa**($\mathcal{N}$=10, $\mathcal{R}$=95%) | 59.81 (-79.0%) | 4.76× | 5158.45 | 5.46× | 7.50 | 5.75 | 6.39 (-13.8%) | 7.52 | 5.77 | 6.41 (-15.0%) |
| **TaylorSeer**($\mathcal{N}$=9, $\mathcal{O}$=2) | 53.92 (-81.1%) | 5.28× | 4515.74 | 6.24× | 6.61 | 6.65 | 6.31 (-14.8%) | 6.67 | 6.63 | 6.31 (-16.3%) |
| **FreqCa**($\mathcal{N}$=9) | **51.09** (-82.0%) | **5.57×** | 4514.48 | 6.24× | **7.62** | **7.18** | **7.27** (-1.9%) | **7.66** | 7.12 | **7.21** (-4.3%) |

- † Methods exhibit significant degradation in image quality; **Q_SC**: semantic consistency, **Q_PQ**: perceptual quality, **Q_O**: overall score.
- Gray: Baseline-relative degradation in quality and gains in latency. Blue: **FreqCa** achieves minimal degradation with large latency gains.

On Qwen-Image-Edit, FreqCa demonstrates superior performance in bilingual editing tasks. At $5.00\times$ speedup, FreqCa achieves Q_O scores of 7.49 on GEdit-CN and 7.52 on GEdit-EN, outperforming TaylorSeer (6.92 and 6.89). At $6.24\times$ speedup, FreqCa shows quality drops of only 1.9% and 4.3%, while TaylorSeer degrades by 14.8% and 16.3%. As shown in Figures 5 and 6 , qualitative evaluation confirms FreqCa's superior visual quality preservation. While FORA ($6.24\times$), Duca($5.46\times$) and TaylorSeer ($6.24\times$) exhibit significant artifacts, FreqCa ($6.24\times$) maintains consistent visual quality comparable to the original model.

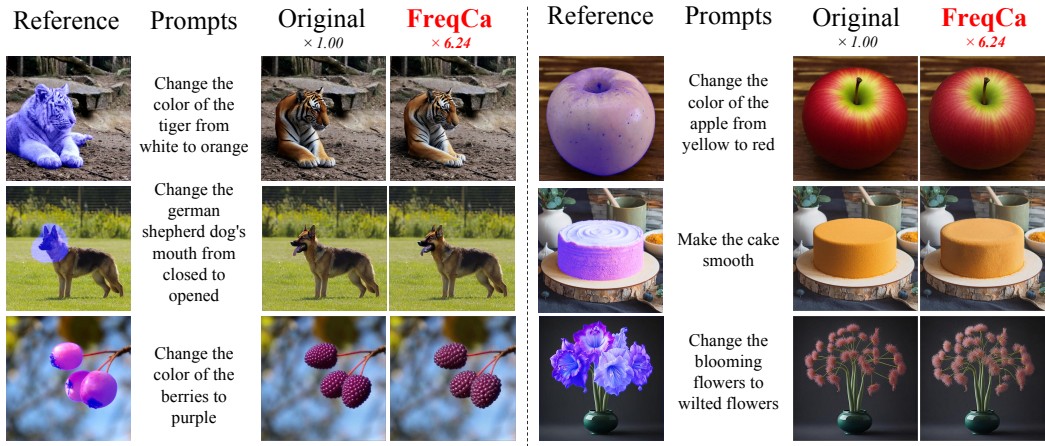

Figure 7: On FLUX.1-Fill-dev, *FreqCa* achieves $6.24\times$ acceleration while preserving image inpainting quality indistinguishable from the original.

### 3.4 ABLATION STUDIES

#### 3.4.1 CACHE MEMORY AND COMPUTATIONAL EFFICIENCY

Conventional layer-wise caching methods store both attention and MLP outputs per layer ($N = 2$) and retain $m+1$ historical states for $m$-th order prediction, yielding memory cost $\mathcal{K}_{\text{layer}} = 2(m+1)L$. For FLUX.1-dev ($L = 57$) with second-order prediction ($m = 2$), this requires 342 cache units.

In contrast, FreqCa caches only the CRF, adopting a frequency-decoupled strategy: low-frequency components are reused (1 unit), while high-frequency components employ second-order Hermite interpolation (3 units). The total cost is constant:

$$\mathcal{K}_{\text{FreqCa}} = 1 + 3 = 4, \quad R = \frac{\mathcal{K}_{\text{FreqCa}}}{\mathcal{K}_{\text{layer}}} = \frac{4}{(m + 1) \cdot N \cdot L} \approx 1.17\% \quad (m = 2, \ L = 57, \ N = 2),$$

reducing memory complexity from $\mathcal{O}(L)$ to $\mathcal{O}(1)$. Computationally, prediction steps incur negligible cost $C_{\text{pred}} \ll C_{\text{full}}$. Executing a full forward pass every $S$ steps yields average cost:

$$\bar{\mathcal{C}} = \tfrac{1}{S}C_{\text{full}} + (1 - \tfrac{1}{S})C_{\text{pred}} \quad \Rightarrow \quad \text{Speedup} \approx S \quad \text{as } C_{\text{pred}} \to 0.$$

FreqCa achieves near-$S\times$ acceleration with only 1% additional memory overhead, establishing the first constant-memory, high-throughput inference acceleration framework for diffusion models.

#### 3.4.2 DECOMPOSITION AND ORDER OF PREDICTION ABLATION STUDY

We perform an ablation study on FLUX.1-dev to identify the optimal frequency decomposition method and prediction order. The study compares three decomposition strategies including FFT, DCT, and a baseline without decomposition, each paired with various prediction approaches for frequency components. Figure 11 compares these optimized configurations. The results indicate that the DCT-based approach, particularly with low-frequency reuse and high-frequency prediction, achieves consistently high ImageReward across acceleration ratios and demonstrates marked superiority at larger intervals ($N > 8$). This robustness under high acceleration factors confirms the rationale for our choice. A separate ablation on Qwen-Image revealed that the FFT-based method with the same prediction strategy performed best. As shown in Figure 8, other configurations result in significant quality degradation compared to our optimal settings.

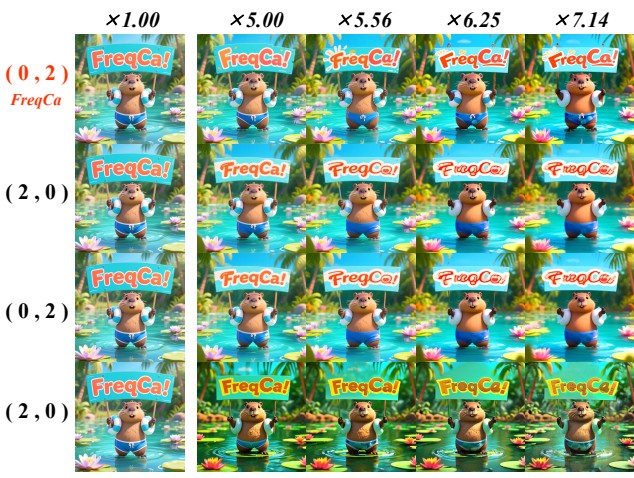
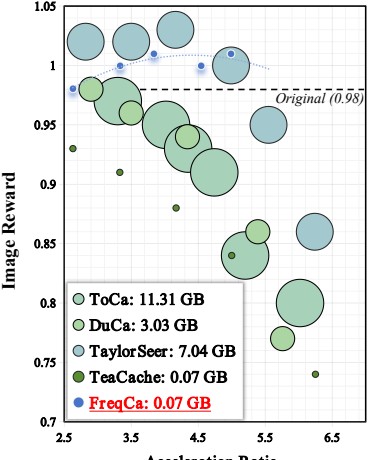

Figure 8: QwenImage ablation results showing image quality under different frequency prediction configurations and acceleration ratios. $(x, y) = (low, high)$ prediction orders.

Figure 9: Imagereward versus speedup ratio across methods. Bubble size indicates cache memory.

Table 5: **Comparison of methods in Cache Memory, MACs, Latency, and FLOPS on FLUX-1.dev**, Best results are highlighted in **bold**, and second-best results are underlined.

| Method | VRAM Overhead (GB)↓ | MACs (T)↓ | Latency (s)↓ | FLOPs (T)↓ | Image Reward↑ |
|---|---|---|---|---|---|
| [dev]: 50 steps | 0.62 | 1859.62 | 23.24 | 3726.87 | 0.99 (+0.0%) |
| **ToCa**($\mathcal{N}$=8, $\mathcal{R}$=75%) | 12.31 (+11.69GB) | 414.88 | 12.39 | 829.86 | 0.95 (-4.0%) |
| **DuCa**($\mathcal{N}$=8, $\mathcal{R}$=70%) | 3.65 (+3.63GB) | 428.86 | 9.40 | 858.27 | 0.94 (-5.1%) |
| **TeaCache**($l$=1.0) | **0.69** (+0.07GB) | 409.43 | 7.07 | 820.55 | 0.84 (-15.2%) |
| **TaylorSeer**($\mathcal{N}$=6, $O$=2) | 7.66 (+7.02GB) | 372.38 | 6.73 | 746.28 | 1.00 (+1.0%) |
| **FreqCa**($\mathcal{N}$=7) | **0.69** (+0.07GB) | **372.25** | **5.19** | **746.03** | **1.01** (+2.0%) |

- **Note:** All methods inherit baseline memory optimizations (e.g., FlashAttention). ToCa is incompatible with these, hence its higher reported cache usage. **Actual cache memory usage** for each method should be computed as: VRAM Overhead − 0.62 GB.

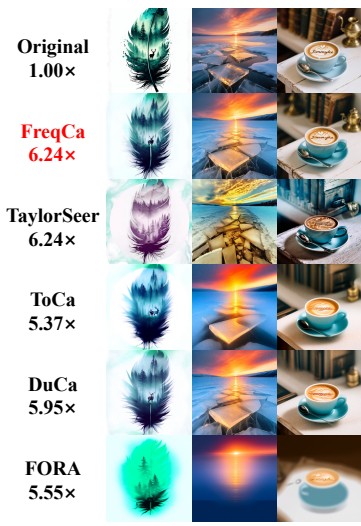
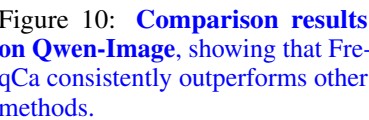
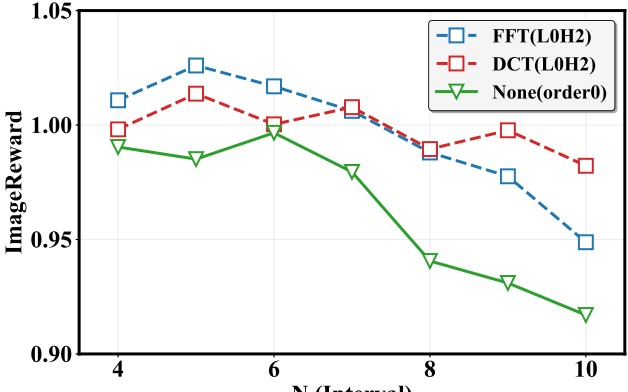

Figure 10: **Comparison results on Qwen-Image,** showing that FreqCa consistently outperforms other methods.

Figure 11: Comparison of optimal predictors under different frequency decomposition methods and prediction strategies on FLUX.1-dev.

## 4 CONCLUSION

In this work, we presented ***FreqCa***, a frequency-aware feature caching framework that unifies the strengths of reuse- and forecast-based paradigms. By decomposing features into low- and high-frequency components, *FreqCa* selectively reuses stable low-frequency features and accurately predicts dynamic high-frequency components, leading to a superior trade-off between acceleration and generation quality. Furthermore, by introducing Cumulative Residual Feature caching, we reduced the memory footprint to $\mathcal{O}(1)$, making frequency-aware caching practical even on consumer hardware. Extensive experiments across diverse diffusion models demonstrate that *FreqCa* achieves $6$–$7\times$ acceleration with negligible quality degradation, establishing a new SOTA in efficient diffusion inference. We believe *FreqCa* opens up new possibilities for scalable, high-performance generative modeling and offers a general method for future research in frequency-aware acceleration techniques.

## ETHICS STATEMENT

This work presents FreqCa, a technical method for accelerating diffusion model inference through frequency-aware caching. Our research focuses purely on computational efficiency improvements and does not introduce new ethical risks beyond those inherent to the underlying diffusion models. We use only publicly available models and datasets in our experiments, and our acceleration technique is model-agnostic and content-neutral. While our method reduces inference time and computational costs, which could potentially make generative AI more accessible, we acknowledge that this accessibility applies to both beneficial and potentially harmful use cases. We encourage responsible deployment of accelerated diffusion models in accordance with existing ethical guidelines for AI-generated content, including proper disclosure of synthetic media and consideration of potential societal impacts.

## REPRODUCIBILITY STATEMENT

We are committed to ensuring the full reproducibility of our FreqCa framework. To this end, Section 3 provides the complete mathematical formulations for our core algorithmic components: frequency decomposition (FFT/DCT), Cumulative Residual Feature (CRF) caching, and second-order Hermite prediction. All experimental configurations are detailed in Section 4.1, specifying the models evaluated (e.g., FLUX.1-dev, Qwen-Image), the datasets used (DrawBench and GEdit), and the full set of evaluation metrics (e.g., ImageReward, CLIP Score, PSNR). Section 4 and the Appendix present our detailed ablation studies, hyperparameter choices for decomposition methods and prediction strategies, and the computational complexity analysis. An anonymous source code repository is provided in the supplementary materials, containing complete inference and training scripts, configuration files with random seeds, and data preprocessing pipelines. The repository will be made publicly available upon acceptance.

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

APPENDIX

## A    USE OF LARGE LANGUAGE MODELS

No Large Language Models were used in this research. All research ideas, algorithmic designs, experimental methodologies, data analysis, and manuscript writing were entirely completed by the authors independently.

## B    RELATED WORKS

Diffusion models have emerged as a cornerstone of modern generative AI, exhibiting state-of-the-art capabilities in synthesizing visual content (Sohl-Dickstein et al., 2015; Ho et al., 2020b). While early models were predominantly built upon U-Net architectures (Ronneberger et al., 2015), their scalability limitations paved the way for the Diffusion Transformer (DiT) (Peebles & Xie, 2023b). The DiT architecture has since become foundational, catalyzing a wave of powerful models across diverse domains (Zheng et al., 2024; Yang et al., 2025). Nevertheless, the iterative nature of the diffusion sampling process imposes a significant computational burden during inference, making acceleration a critical area of research (Ho et al., 2020b; Peebles & Xie, 2023b). Current efforts to enhance efficiency are largely focused on two complementary directions: reducing the number of sampling steps and accelerating the denoising network itself.

### B.1    SAMPLING TIMESTEP REDUCTION

One primary strategy seeks to minimize the number of required sampling iterations while preserving generation quality. Seminal work like DDIM introduced deterministic sampling to reduce step counts without significant fidelity loss (Song et al., 2021). This concept was further refined by the DPM-Solver series, which employed high-order ODE solvers to achieve faster convergence (Lu et al., 2022a;b; Zheng et al., 2023). Other notable approaches include knowledge distillation, which trains a student model to emulate multiple denoising steps of a larger teacher model (Salimans & Ho, 2022; Meng et al., 2022), and Rectified Flow, which learns to straighten the generation path between noise and data distributions (Liu et al., 2023b). More recently, Consistency Models have enabled high-quality synthesis in a single step by directly mapping noise to clean data, circumventing the need for a sequential path (Song et al., 2023).

### B.2    DENOISING NETWORK ACCELERATION

An alternative to reducing timesteps is to decrease the computational cost of each forward pass through the denoising network. This is typically achieved via model compression or feature caching.

**Model Compression-based Acceleration.**    One avenue involves model compression, which includes techniques such as network pruning (Fang et al., 2023; Zhu et al., 2024), quantization (Li et al., 2023b; Shang et al., 2023; Kim et al., 2025), and various forms of token reduction that dynamically shorten the input sequence length (Bolya & Hoffman, 2023; Kim et al., 2024; Zhang et al., 2024; 2025). While effective, these methods often necessitate a fine-tuning or retraining stage to mitigate the potential loss of expressive power inherent in model simplification (Li et al., 2024; 2023b).

**Feature Caching-based Acceleration.**    A compelling training-free alternative is feature caching, which exploits temporal redundancies in the denoising process. Pioneered in U-Net architectures through FasterDiffusion and DeepCache, this paradigm was subsequently adapted to DiTs. Initial efforts focused on a "cache then reuse" strategy, while advanced techniques like FORA and $\Delta$-DiT refined this approach. This concept evolved with more sophisticated mechanisms, including dynamic token-level updates (ToCa), adaptive sampling (RAS (Liu et al., 2025c)), and explicit error correction frameworks (Qiu et al., 2025; Chen et al., 2025; Chu et al., 2025). A pivotal shift was the "cache then forecast" paradigm introduced by TaylorSeeer, which was further advanced by more robust numerical methods in FoCa (Zheng et al., 2025), HiCache (Feng et al., 2025), and SpeCa (Liu et al., 2025b). OmniCache (Chu et al., 2025) primarily addresses the temporal scheduling of cache reuse based on trajectory curvature, employing filtering solely for noise correction rather than feature decoupling.

However, a crucial flaw underlies these sophisticated paradigms, as hinted at by preliminary frequency-domain analyses. For instance, FreeU (Si et al., 2023) enhances high-frequency structural details by statically rebalancing skip-connection and backbone features within the U-Net architecture. PAB (Zhao et al., 2024) insightfully associated different attention mechanisms with distinct frequency bands but did not delve into token-level frequency dynamics. Similarly, while FasterCache (Lv et al., 2025) examined the frequency-domain differences within Classifier-Free Guidance , its findings were confined to this specific context, not addressing the more universal dynamics of temporal feature evolution and thus showing limited practical acceleration.

In contrast to prior methods that treat features as a monolithic whole, we propose **FreqCa**, which resolves quality degradation in caching by decomposing features into their stable low-frequency and volatile high-frequency components for differentiated treatment. As an added benefit, we introduce the Cumulative Residual Feature, collapsing the memory complexity from $\mathcal{O}(L)$ to $\mathcal{O}(1)$ to solve the resource inefficiency of prior "layer-wise" architectures.

## C   DETAILED EXPERIMENTAL SETUP

This section provides comprehensive technical details for all experimental configurations.

### C.1   MODEL AND TASK SPECIFICATIONS

**FLUX.1-dev and Qwen-Image:**   The images generated by the FLUX.1-dev, FLUX.1-schnell, and Qwen-Image-Lightning models are obtained at 1024x1024 resolution using 200 high-quality prompts sourced from the DrawBench benchmark, while those generated by Qwen-Image were obtained at $1328 \times 1328$ resolutions. Quality assessment is performed using ImageReward, a robust perceptual metric for text-to-image alignment.

**FLUX.1-Kontext-dev and Qwen-Image-Edit:**   We employ the FLUX.1-Kontext-dev and Qwen-Image-Edit model for image editing synthesis. Images editing and quality assessment is performed using GEdit benchmark, which grounded in real-world usages is developed to support more authentic and comprehensive evaluation of image editing models.

### C.2   HARDWARE AND COMPUTATIONAL RESOURCES

All experiments are conducted on enterprise-grade GPU infrastructure:
- FLUX.1-dev experiments: NVIDIA A100 GPU
- FLUX.1-Kontext-dev experiments: NVIDIA A100 GPU
- Qwen-Image experiments: NVIDIA H20 GPU
- Qwen-Image-Edit experiments: NVIDIA H20 GPU

### C.3   FREQCA IMPLEMENTATION PARAMETERS

- FLUX.1-dev experiments: DCT-based frequency decomposition was adopted.
- FLUX.1-Kontext-dev experiments: DCT-based frequency decomposition was adopted.
- Qwen-Image experiments: FFT-based frequency decomposition was adopted.
- Qwen-Image-Edit experiments: FFT-based frequency decomposition was adopted.

## D   DECOMPOSITION AND ORDER OF PREDICTION ABLATION STUDY

As shown in Figure D1, we systematically compared classical frequency decomposition methods (FFT and DCT) with a baseline that does not perform any frequency decomposition. The results clearly demonstrate that frequency decomposition plays a critical role in stabilizing model performance: omitting decomposition leads to a sharp drop in ImageReward scores, whereas both FFT and DCT significantly mitigate this degradation and maintain stable quality across timesteps. Furthermore, we investigated the impact of different prediction orders for low- and high-frequency components.

We observe that inappropriate prediction strategies can easily introduce errors and harm generation quality. Among all tested configurations, the combination of zeroth-order prediction (direct reuse) for low-frequency components and second-order prediction for high-frequency components achieves consistently superior results, validating our hypothesis that low-frequency features should be reused directly while high-frequency components benefit from higher-order modeling. These findings confirm the necessity of frequency-aware design and provide empirical guidance for selecting optimal prediction strategies.

## D.1 PREDICTION ORDER COMBINATIONS

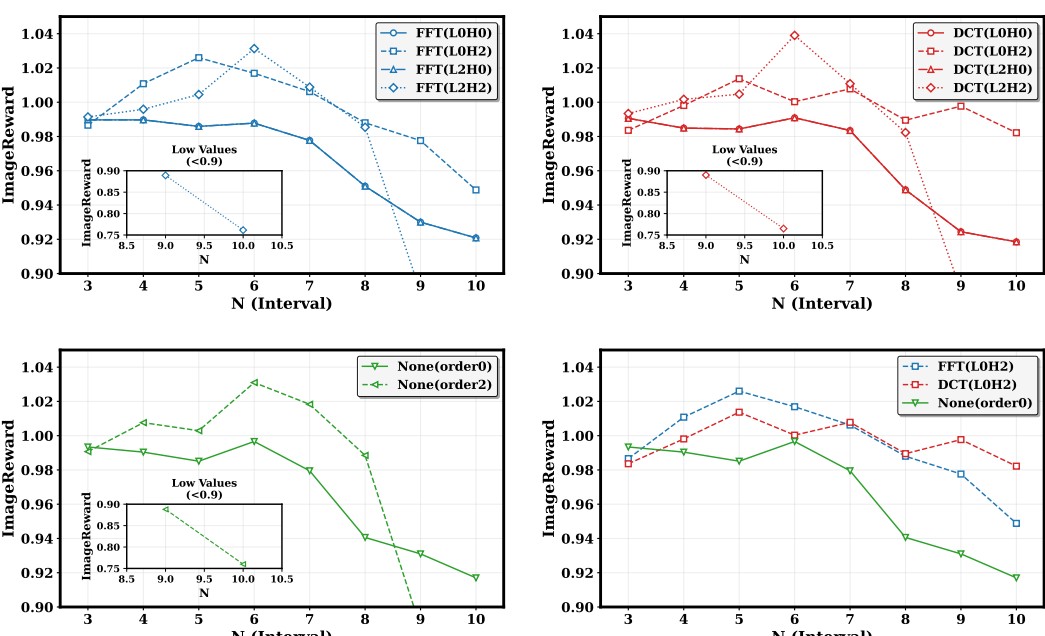

Figure D1: The ImageReward scores of FLUX.1-dev's decomposition strategies (FFT, DCT, no-decomposition) paired with different frequency prediction approaches are presented here. This content includes the optimal prediction method for each decomposition strategy where low-frequency reuse and high-frequency prediction apply to FFT and DCT, and direct reuse applies to the no-decomposition (None) strategy.

## E    APPLICABILITY OF FREQCA TO U-NET ARCHITECTURES

We have successfully validated FreqCa on the U-Net-based Stable Diffusion XL (SDXL) to confirm its generalization. Comparative experiments against DeepCache demonstrate that FreqCa consistently achieves superior semantic alignment (ImageReward) and perceptual fidelity (LPIPS) across all acceleration intervals. Specifically, at a standard interval of $N = 2$, FreqCa achieves an ImageReward of 0.4719 and LPIPS of 0.2080, surpassing DeepCache .

Crucially, FreqCa exhibits remarkable robustness at high acceleration ratios: at $N = 4$, while DeepCache suffers severe degradation (ImageReward drops to 0.3255), FreqCa maintains high quality with an ImageReward of 0.4183, significantly outperforming the baseline. These results verify that FreqCa is architecture-agnostic, as it exploits the intrinsic frequency dynamics of feature activations common to both Transformers and U-Nets.

Table E1: Performance comparison across different skip intervals using ImageReward and LPIPS.

| Method | Interval ($N$) | ImageReward ↑ | LPIPS ↓ |
|---|---|---|---|
| Original | 1 | 0.5157 | – |
| DeepCache | 2 | 0.4484 | 0.2582 |
| **FreqCa** | 2 | 0.4719 | 0.2080 |
| DeepCache | 3 | 0.4219 | 0.3104 |
| **FreqCa** | 3 | 0.4721 | 0.2693 |
| DeepCache | 4 | 0.3255 | 0.3591 |
| **FreqCa** | 4 | 0.4183 | 0.2946 |

## F    APPLICABILITY OF FREQCA TO VIDEO DIFFUSION MODELS

We conducted additional validation on HunyuanVideo, evaluating performance under a challenging setting of 480p resolution, 65 frames, and a high acceleration ratio of **5.56**×. As shown in Table F2, FreqCa achieves state-of-the-art performance on the VBench metric, matching the quality of the memory-intensive TaylorSeer while significantly outperforming TeaCache. Importantly, FreqCa

Table F2: VBench comparison on HunyuanVideo under $5.56\times$ acceleration.

| Method | VBench |
|---|---|
| TeaCache | 78.32 |
| TaylorSeer | 79.43 |
| **FreqCa** | **79.40** |

reaches this high performance with minimal memory overhead. TaylorSeer caches full feature maps at every layer, resulting in extremely high memory usage. In contrast, FreqCa caches only the **Cumulative Residual Feature (CRF)**, leading to a cache footprint of **less than 1% of TaylorSeer's** while achieving comparable VBench scores. The extreme memory efficiency of FreqCa is not merely a theoretical benefit; it substantially broadens the range of feasible acceleration scenarios. Whereas the large caching cost of methods such as TaylorSeer often results in **Out-Of-Memory (OOM)** failures in high-resolution or long-sequence settings (e.g., 720p with 120+ frames), FreqCa's lightweight caching enables it to **support these high-definition, long-horizon video generation tasks** without difficulty. This demonstrates that FreqCa is currently the **only practical acceleration solution** for computationally intensive, high-fidelity video synthesis.

# G    SENSITIVITY TO FFT/DCT CHOICE AND FREQUENCY SPLIT THRESHOLD

Our analysis shows that the **feature spectrum naturally exhibits a clear bimodal distribution**, meaning that the low- and high-frequency components are already well-separated without a substantial intermediate band. Low-frequency components encode coarse, static structural information, whereas high-frequency components capture fine and rapidly varying details. This inherent spectral separation is the primary reason why FreqCa is **robust to the precise choice of the cutoff threshold**. As demonstrated in the ablation study below, varying the cutoff ratio over a wide range (0.1 to 0.4) results in only minor fluctuations in performance. The results indicate that once the ratio is set

Table G3: Ablation of cutoff ratio for frequency separation.

| Cutoff Ratio | ClipScore | ImageReward | LPIPS |
|:---:|:---:|:---:|:---:|
| **0.05** | 34.5414 | 0.9411 | 0.4289 |
| **0.1** | 34.8449 | 1.0391 | 0.4053 |
| **0.2** | 35.0517 | 1.0420 | 0.3925 |
| **0.3** | 35.0635 | 1.0484 | 0.3943 |
| **0.4** | 35.2631 | 1.0615 | 0.3976 |

beyond a minimal threshold (e.g., $\geq 0.1$), the performance metrics remain stable, confirming the insensitivity of the method. The only notable degradation occurs when the ratio is set too low (e.g., 0.05), where essential low-frequency information is incorrectly mixed into the high-frequency band, leading to a loss of structural fidelity. Furthermore, we explored more sophisticated decomposition strategies, including **adaptive frequency partitioning** and **multi-band splitting**. However, the additional performance gains over a fixed ratio were marginal. This suggests that the complexity of adaptive or multi-band decomposition is unnecessary, precisely because the natural feature spectrum is already sufficiently separated. Consequently, the simple fixed-ratio split offers an optimal balance among robustness, performance, and computational simplicity.

# H    DISCUSSION ON CUMULATIVE RESIDUAL FEATURES (CRF) FIDELITY UNDER DEEP AND NON-RESIDUAL ARCHITECTURES

## H.1    ROBUSTNESS ON ULTRA-DEEP ARCHITECTURES

To examine sensitivity to network depth, we evaluate FreqCa on several extremely deep diffusion backbones, including **Flux-dev (57 transformer blocks)** and **Qwen-Image (60 layers)**. Across these large models, FreqCa consistently maintains stable generation quality, demonstrating that frequency-aware prediction effectively prevents error accumulation. Additional validation on **U-Net architectures** further shows that this robustness extends across architectures with fundamentally different signal propagation dynamics, indicating that FreqCa's effectiveness is not dependent on specific residual pathways.

## H.2    TRADE-OFF ANALYSIS ON VIDEO DIFFUSION MODELS

We further tested CRF on the more demanding **HunyuanVideo** model, which features large parameter counts and significant temporal depth. While **TaylorSeer** benefits from caching full features at every layer and therefore retains more raw information, this comes with extremely large memory overhead. In contrast, FreqCa achieves a **VBench score of 79.40**, nearly identical to TaylorSeer (79.43), while requiring **less than 1% of its cache memory**. This demonstrates that although CRF is a compressed representation, FreqCa's frequency-aware compensation achieves an optimal balance between fidelity and resource efficiency for deep and memory-intensive video generation settings.

# I    IMPLEMENTATION DETAILS AND ALGORITHMIC ANALYSIS OF FREQCA

## I.1    CACHING STRATEGY AND INTERACTION WITH HYPERPARAMETER $N$

The hyperparameter $N$ serves as the core control for the **caching and reuse strategy** within FreqCa. To address the instability of predicted noise in the early stages of multi-step diffusion, we implement a `first_enhance` stage, where the initial few steps are computed entirely by the main model. In subsequent stages, the model operates based on the interval $N$: every $N - 1$ steps, the main model performs a **full computation step** (denoted as C). The intervening steps are **prediction steps** (denoted as P), handled exclusively by the FreqCa predictor without executing the diffusion backbone.

For example, in a 10-step diffusion process with `first_enhance = 2` and an interval $N = 3$, the execution sequence is `CCPPCPPCPP`. During prediction steps, the diffusion model is bypassed. Instead, we reuse the low-frequency component from the most recent full computation step and synthesize the high-frequency component using a Hermite-based predictor.

**Full Computation Steps (Generation Logic):** At these steps, we decompose the model output into frequency bands and compute finite-difference residuals for the high-frequency data. Let $F_0, F_1, F_2$ be the high-frequency data from the latest three full computation steps at times $t_0, t_1, t_2$. The specific difference calculations are:

$$\Delta F_{curr}^{(1)} = \frac{F_0 - F_1}{t_0 - t_1}$$

$$\Delta F_{curr}^{(2)} = \frac{\Delta F_{curr}^{(1)} - \Delta F_{prev}^{(1)}}{t_0 - t_1} = \frac{\frac{F_0 - F_1}{t_0 - t_1} - \frac{F_1 - F_2}{t_1 - t_2}}{t_0 - t_1}$$

**Prediction Steps (Usage Logic):** We reuse the low-frequency data from the previous full computation step and predict the high-frequency data $\widehat{F}_{\text{high}}$ as follows:

$$\widehat{F}_{\text{high}}(d) = F_0 + \alpha_1(d)\Delta F^{(1)} + \alpha_2(d)\Delta F^{(2)}$$

Here, $F_0$ is the high-frequency data from the previous full computation step, and $\Delta F^{(1)}, \Delta F^{(2)}$ are the derivatives computed at that step. The coefficients $\alpha_k(d)$ are derived from Hermite polynomials:

$$\alpha_k(d) = \frac{H_k(x)}{k!}s^k, \quad H_1(x) = 2x, \quad H_2(x) = 4x^2 - 2, \quad x = sd$$

where $s$ is a tuning factor and $d$ is the distance to the previous full computation step. The final output is the sum of the cached low-frequency data and the predicted high-frequency data.

## I.2    SPECTRAL FEATURE DECOMPOSITION DETAILS

In our diffusion transformer, latent features $X_{\text{tok}} \in \mathbb{R}^{B \times L \times D}$ are reshaped into spatial feature maps $X \in \mathbb{R}^{B \times C \times H \times W}$ via the transformation $[B, L, D] \to [B, H, W, D] \to [B, C, H, W]$. We utilize a **2D Type-II Discrete Cosine Transform (DCT)** rather than FFT to avoid boundary ringing artifacts caused by periodic extension. The transform is defined as:

$$Y_{b,c}(u, v) = \alpha_u \alpha_v \sum_{x=0}^{H-1} \sum_{y=0}^{W-1} X_{b,c}(x, y) \cos\left[\frac{\pi}{H}\left(x + \tfrac{1}{2}\right)u\right] \cos\left[\frac{\pi}{W}\left(y + \tfrac{1}{2}\right)v\right]$$

Using a radial mask with cutoff $\tau$, we separate the spectrum into $Y_{\text{low}}$ and $Y_{\text{high}}$. By linearity of the inverse transform $\mathcal{T}^{-1}$, we obtain the decomposed features:

$$X_{\text{low}} = \mathcal{T}^{-1}(Y \odot M_{\text{low}}), \quad X_{\text{high}} = \mathcal{T}^{-1}(Y \odot M_{\text{high}})$$

$X_{\text{low}}$ represents the stable structural component suitable for caching, while $X_{\text{high}}$ captures volatile details handled by the predictor. For implementation, we use `torch_dct.dct_2d` with orthogonal normalization.

### I.3 SECOND-ORDER HERMITE PREDICTOR AND EFFICIENT COMPUTATION

To evaluate the trajectory efficiently, we employ a forward-difference operator $\Delta g(d) = g(d+1) - g(d)$. The second-order Hermite interpolator used in FreqCa corresponds to:

$$\widehat{F}(d) = F_0 + (2s^2 d)\Delta F^{(1)} + (2s^4 d^2 - s^2)\Delta F^{(2)}$$

Since $\widehat{F}(d)$ is a quadratic polynomial in $d$, all differences of order $m \geq 3$ vanish. We can thus simplify the computation using recursive forward differences:

$$\Delta \widehat{F}(d) = 2s^2 \Delta F^{(1)} + (4s^4 d + 2s^4)\Delta F^{(2)}$$

$$\Delta^2 \widehat{F}(d) = 4s^4 \Delta F^{(2)}, \qquad \Delta^m \widehat{F}(d) = 0 \quad \text{for } m \geq 3$$

In implementation, $\Delta^2 \widehat{F}(d)$ is a constant computed once per interval, while $\Delta \widehat{F}(d)$ is updated recursively. This closed-form scheme ensures the predictor's overhead remains negligible while accurately modeling the high-frequency dynamics.

## J STABILITY UNDER HIGH ACCELERATION RATIOS

### J.1 STABILITY ANALYSIS AND PRACTICAL LIMITS

Empirically, this theoretical advantage yields strong stability across a wide range of acceleration ratios. Our experiments, covering intervals from $N = 3$ to $16$, demonstrate a broad effective operating range:

- **Stable Regime** ($N \leq 10$)**:** The method remains robust up to $N = 10$ ($\approx 6.2\times$ speedup). Key metrics remain close to the baseline (e.g., ImageReward $\approx 0.98$–$1.01$, ClipScore $\approx 32.6$).
- **Physical Limit** ($N \geq 12$)**:** Significant degradation emerges beyond $N \geq 12$, where the diffusion trajectory becomes too sparse for reliable reconstruction by *any* predictor. Thus, the performance drop reflects the fundamental sampling limit rather than predictor instability.

Table J4: Stability of the Hermite predictor across different sampling intervals $N$.

| Interval | Speed | ClipScore | ImageReward | LPIPS |
|---|---|---|---|---|
| **3** | 2.63 | 32.61 | 0.98 | 0.13 |
| **4** | 3.33 | 32.70 | 1.00 | 0.19 |
| **5** | 3.84 | 32.76 | 1.01 | 0.23 |
| **6** | 4.54 | 35.26 | 1.00 | 0.28 |
| **7** | 4.99 | 32.89 | 1.01 | 0.32 |
| **8** | 5.54 | 32.71 | 0.99 | 0.36 |
| **9** | 5.54 | 32.67 | 1.00 | 0.38 |
| **10** | 6.23 | 32.56 | 0.98 | 0.43 |
| **12**[†] | 7.12 | 32.42 | 0.89 | 0.51 |
| **16**[†] | 8.31 | 30.94 | 0.61 | 0.62 |

[†] Indicates severe image quality collapse at large sampling intervals.

## K ORDER SELECTION AND HYPERPARAMETER STUDY

To maximize performance in the stable regime, we tune the predictor order to balance modeling capability and memory overhead. A first-order predictor (linear assumption) is memory-efficient but

inadequate for nonlinear diffusion trajectories. The **second-order predictor** captures curvature with only a marginal increase in memory cost to store derivative states. Under the challenging setting of $N = 10$, the second-order Hermite predictor substantially outperforms the first-order version across all quality metrics:

Table K5: Comparison of Hermite predictor orders under interval $N = 10$.

| Method | Interval ($N$) | ClipScore | ImageReward | PSNR | SSIM | LPIPS |
|---|---|---|---|---|---|---|
| **Hermite** ($O = 1$) | 10 | 34.6829 | 0.9877 | 28.903 | 0.6550 | 0.4165 |
| **Hermite** ($O = 2$) | 10 | 34.8449 | 1.0391 | 28.953 | 0.6654 | 0.4053 |

These findings identify the second-order Hermite predictor as the optimal configuration, delivering improved reconstruction quality and stability with minimal overhead.

## L    COMPARISON ON FEW-STEP DIFFUSION MODELS

### L.1    EVALUATION ON QWEN-IMAGE-LIGHTNING (8 STEPS)

To evaluate performance under stringent step constraints, we conduct a comparative study against TaylorSeer using **Qwen-Image-Lightning (8 steps)**, the state-of-the-art distilled few-step diffusion model. Results show that FreqCa consistently outperforms the Taylor-based baseline in both semantic alignment and perceptual fidelity. Notably, FreqCa demonstrates strong robustness at high acceleration ratios. At an interval of $N = 4$, TaylorSeer suffers significant degradation (ImageReward drops to 1.20), whereas **FreqCa maintains an ImageReward of** 1.29, matching the original Lightning model exactly. In terms of perceptual quality, FreqCa achieves a substantially lower LPIPS of **0.12** at $N = 2$, compared with TaylorSeer's 0.28. These results confirm that FreqCa's frequency-aware prediction preserves critical feature dynamics more effectively than Taylor-based extrapolation, making it particularly well-suited for few-step generation where each step must be maximally informative.

Table L6: Comparison with TaylorSeer on Qwen-Image-Lightning (8-step distilled model).

| Method | Interval | ImageReward | LPIPS |
|---|---|---|---|
| Lightning (8 steps) | 1 | 1.29 | – |
| TaylorSeer | 2 | 1.27 | 0.28 |
| **FreqCa** | 2 | **1.29** | **0.12** |
| TaylorSeer | 3 | 1.24 | 0.35 |
| **FreqCa** | 3 | **1.28** | **0.18** |
| TaylorSeer | 4 | 1.20 | 0.42 |
| **FreqCa** | 4 | **1.29** | **0.36** |

### L.2    EVALUATION UNDER DIRECT STEP COMPRESSION (50 → 20 STEPS)

Since Qwen-Image maintains stable generation quality at the original 50-step inference setting, we directly compress it to 20 steps as the baseline. Its ImageReward remains high, indicating that Qwen-Image is relatively robust to reduced-step inference. Thus, using the 20-step version as the baseline is both reasonable and representative. Under the 20-step setting, we keep the same cache interval as in the main experiments. In this scenario, TaylorSeer exhibits a clear collapse in performance and fails to deliver the expected acceleration benefits. We believe this is because **directly compressing the number of inference steps is a crude approximation**, which disrupts the

model's structural representation of high-frequency and low-frequency components more severely than distillation-based methods. TaylorSeer applies uniform processing to all features without distinguishing frequency components, amplifying the distribution shift caused by step compression and leading to significant degradation. In contrast, our proposed FreqCa explicitly separates and models high- and low-frequency features. This frequency-aware design remains robust even under aggressively reduced steps, preserving strong image quality. This demonstrates that FreqCa is more resilient to feature-distribution shifts and better suited for scenarios where aggressive accelerations (e.g., $5\times$–$7\times$) are required.

Table L7: Performance under direct step compression from 50 to 20 steps.

| Method | Interval | ImageReward | LPIPS |
|---|---|---|---|
| Qwen-Image (50 steps → 20 steps) | 1 | 1.19 | 0 |
| TaylorSeer | 5 | -1.16 | 0.61 |
| **FreqCa** | 5 | **1.09** | **0.22** |

## M    THEORETICAL ANALYSIS ON ERROR BOUNDS

Thank you for the question. We clarify that the main contribution of FreqCa does not lie in selecting a particular predictor, but rather in introducing a **frequency-aware decomposition** that separates feature activations into low- and high-frequency components before extrapolation. This design allows these components to be handled by appropriate predictors, rather than applying a single extrapolator to the entire feature space as done in TaylorSeer. In this sense, the predictor itself is largely orthogonal to the performance gain; the improvement primarily stems from **predicting different frequency bands separately**, rather than from any specific extrapolation formula. Nevertheless, we evaluated several candidate predictors for the high-frequency branch, including Hermite, Taylor, Chebyshev, and Laguerre interpolation. Across all settings, **second-order Hermite interpolation consistently provides the best stability and perceptual quality**. Hermite benefits from using both function values and first-order derivatives, enabling it to preserve local curvature and maintain shape continuity—properties particularly important for high-frequency components, which exhibit rapid temporal variations and are more sensitive to extrapolation errors. In contrast, Taylor expansion tends to accumulate error quickly under nonlinear diffusion dynamics, whereas Chebyshev and Laguerre polynomials introduce oscillatory behaviors that reduce robustness. Our experiments, summarized in Table M8, show that at the same acceleration ratio ($7.14\times$), Hermite achieves an ImageReward of 1.04, outperforming Taylor (0.98), Laguerre (0.97), and Chebyshev (0.95). Moreover, applying Taylor directly on the full features, as in TaylorSeer, yields substantially worse results (0.73), further highlighting the importance of frequency decoupling combined with a stable high-frequency predictor. These findings confirm that while the frequency decomposition is the core novelty of our method, Hermite is the most reliable choice for predicting the high-frequency component at high acceleration ratios.

Table M8: Comparison of different high-frequency predictors under similar acceleration ratios.

| Method | Predictor | Speed | ImageReward |
|---|---|---|---|
| TaylorSeer | Taylor | 6.25 | 0.73 |
| **FreqCa** | Hermite | 7.14 | **1.04** |
| **FreqCa** | Taylor | 7.14 | 0.98 |
| **FreqCa** | Chebyshev | 7.14 | 0.95 |
| **FreqCa** | Laguerre | 7.14 | 0.97 |

