# OpenReview forum: "FreqCa: Accelerating Diffusion Models via Frequency Decomposition Caching"
_ICLR.cc/2026/Conference — ICLR 2026 Conference Desk Rejected Submission_

### Official Review · Reviewer_wkDH · 2025-10-29

**Soundness:** 2
**Presentation:** 2
**Contribution:** 2
**Rating:** 4
**Confidence:** 4

**Summary:**

This paper proposes a frequency-aware caching framework called FreqCa to accelerate the sampling efficiency of diffusion models. FreqCa decomposes cached features into low-frequency and high-frequency components, then reuses the low-frequency part and predicts the high-frequency part using historical high-frequency information and a Hermite interpolator. Additionally, this paper proposes to cache only the Cumulative Residual Features (CRF), which significantly reduces memory overhead. Experimental results show that FreqCa consistently demonstrates strong competitiveness in terms of speed and memory usage in benchmark tests of text-to-image generation and image editing.

**Strengths:**

1. The method is based on a sound motivation. Figure 2 illustrates the different behaviors of low-frequency and high-frequency components during the diffusion model sampling process, proving the rationality and effectiveness of the differentiated caching strategy for low and high frequencies.
2. FreqCa combines differentiated frequency processing with memory-efficient CRF caching, outperforming previous methods that cache entire features.
3. The experimental results are sufficient. Extensive experiments are conducted on different models and tasks, fully verifying the effectiveness of FreqCa.

**Weaknesses:**

1. The different behaviors of low-frequency and high-frequency components during diffusion model sampling have already been discovered in previous work (FreeU [1]), although that study focused on UNet models.
2. The paper lacks elaboration on the implementation of its methods. For example, it does not explain how FFT and DCT work in detail, nor how the m-th order difference $\Delta^{m}F(z_{t}^{High})$ of the second-order Hermite interpolator is calculated.
3. Although experimental results show that Hermite interpolation works well, there is a lack of theoretical analysis on error bounds or convergence, as well as comparisons with other interpolation methods.

[1] Si C, Huang Z, Jiang Y, et al. Freeu: Free lunch in diffusion u-net[C]//Proceedings of the IEEE/CVF Conference on Computer Vision and Pattern Recognition. 2024: 4733-4743.

**Questions:**

1. In Figure 3(d), why are $\Delta^{1}F(z_{t}^{High})$ and $\Delta^{2}F(z_{t}^{High})$ connected bidirectionally to $\Delta^{1}F(z_{t+N}^{High})$ and $\Delta^{2}F(z_{t+N}^{High})$ respectively? Do these two pairs of components influence each other?
2. Why is a second-order Hermite interpolator chosen to predict the volatile high-frequency components? Have different methods been tried for predicting high-frequency components? How do their performances compare to the second-order Hermite interpolator, and are there specific comparison results?
3. What is the maximum acceleration multiple of FreqCa? Is there an acceleration threshold beyond which the quality of generated images degrades sharply?
4. Are there special cases where FreqCa fails to achieve acceleration? If yes, can the reasons for this failure be explained?

---

> ### Author Response · Authors · 2025-11-20
> **Response to Q1: Relation to FreeU and prior observations on low- vs. high-frequency behavior**
>
> We appreciate the comment. While FreeU shares the observation regarding the distinct temporal dynamics of frequency bands, its methodology is **orthogonal** to ours. FreeU is a **quality enhancement technique** that implicitly rebalances feature contributions via static scaling factors. It neither performs actual spectral analysis nor reduces sampling cost.
>
> In contrast, FreqCa leverages these dynamics for **sampling acceleration**. Our core contribution is the **explicit decomposition** of features using FFT/DCT, combined with a predictive caching mechanism (Reuse + Hermite). This allows us to skip computational steps while maintaining fidelity. Thus, our work extends the general frequency observation into a novel domain—efficient feature extrapolation. **We will include a detailed discussion and comparative analysis with FreeU in the final version to clarify these fundamental differences.**

---

> ### Author Response · Authors · 2025-11-20
> **Response to Q2:  Acceleration Threshold and Boundary Conditions**
>
> Our extensive evaluation defines the operational limits and boundary behaviors of FreqCa. regarding acceleration limits, FreqCa exhibits a broad stable regime, maintaining high generation quality up to an interval of **$N=10$** (corresponding to a **$6.23\times$ speedup**), where semantic metrics remain robust (ImageReward $\approx 0.98$). Sharp degradation appears only at **$N \ge 12$** ($\approx 7.12\times$ speedup), marking the fundamental limit of sparse trajectory sampling rather than a method-specific failure. Regarding boundary cases, we examined memory-intensive **Video Generation** as a stress test. While FreqCa achieves a Vbench score of **79.40**, which is negligibly lower than the memory-heavy TaylorSeer (79.43) but significantly higher than TeaCache (78.32), this slight margin highlights a critical advantage: FreqCa delivers this near-SOTA performance with a **cache cost of less than 1% of TaylorSeer’s**. Thus, what appears to be a performance boundary is actually a demonstration of extreme efficiency, enabling FreqCa to support high-resolution video tasks that cause OOM errors in other baselines.
>
> | **Interval** | **speed** | ClipScore | ImageReward | LPIPS |
> | --- | --- | --- | --- | --- |
> | **3** | 2.63 | 32.6116 | 0.9836 | 0.131 |
> | **4** | 3.33 | 32.6989 | 0.9981 | 0.1924 |
> | **5** | 3.84 | 32.7552 | 1.0137 | 0.2287 |
> | **6** | 4.54 | 35.2631 | 1.0003 | 0.2791 |
> | **7** | 4.99 | 32.8939 | 1.0078 | 0.3152 |
> | **8** | 5.54 | 32.7057 | 0.9895 | 0.362 |
> | **9** | 5.54 | 32.6725 | 0.9977 | 0.3773 |
> | **10** | 6.23 | 32.5634 | 0.9822 | 0.4298 |
> | **12** | 7.12 | 32.4238 | 0.8911 | 0.5141 |
> | **16** | 8.31 | 30.9423 | 0.6072 | 0.616 |

---

> ### Author Response · Authors · 2025-11-20
> **Response to Q3: Theoretical Analysis on Error Bounds**
>
> Thank you for the question. We would like to clarify that the main contribution of FreqCa does not lie in selecting a particular predictor, but in introducing a **frequency-aware decomposition** that separates feature activations into low- and high-frequency components before extrapolation. This design allows these componentsn to be handled by appropriate predictors, rather than applying a single extrapolator to the entire feature space as in TaylorSeer. In this sense, the predictor itself is largely orthogonal to the performance gain; the improvement primarily comes from **predicting different frequency bands separately**, rather than from any specific formula used for extrapolation.
>
> Nevertheless, we evaluated several candidate predictors for the high-frequency branch, including Hermite, Taylor, Chebyshev, and Laguerre interpolation. Across all settings, **second-order Hermite interpolation consistently offered the best stability and perceptual quality**. Hermite benefits from using both function values and first-order derivatives, enabling it to preserve local curvature and maintain shape continuity—properties particularly important for high-frequency components, which exhibit rapid temporal variations and are more sensitive to extrapolation errors. In contrast, Taylor expansion tends to accumulate error quickly under nonlinear diffusion dynamics, while Chebyshev and Laguerre polynomials introduce oscillatory behaviors that reduce robustness. Our experiments, summarized in the table below, show that at the same acceleration ratio (7.14×), Hermite achieves an ImageReward of 1.04, outperforming Taylor (0.98), Laguerre (0.97), and Chebyshev (0.95). Moreover, applying Taylor directly on the full features, as in TaylorSeer, yields substantially worse results (0.73), further highlighting the importance of frequency decoupling combined with a stable high-frequency predictor. These results confirm that while the frequency decomposition is the core novelty, Hermite is the most reliable choice for predicting the high-frequency component at high acceleration ratios.
>
> | **method** | **predictor** | Speed | ImageReward |
> | --- | --- | --- | --- |
> | **FreqCa** | Hermite | 7.14 | 1.04 |
> | **FreqCa** | taylor | 7.14 | 0.98 |
> | **FreqCa** | chebyshev | 7.14 | 0.95 |
> | **FreqCa** | laguerre | 7.14 | 0.97 |
> | TaylorSeer | taylor | 6.25 | 0.73 |

---

> ### Author Response · Authors · 2025-11-20
> **Response to Q4: Connectivity and Recursive Computation in Figure 3(d)**
>
> Thank you for the question. We would like to kindly clarify that the connections in Figure 3(d) do not imply bidirectional dynamic influence. Instead, they illustrate the **computational dependency flow** required for the **recursive calculation of high-order finite differences**.
>
> ---
>
>  **1. Recursive Finite Difference Calculation**
>
> To estimate the $m$-th order derivative at timestep $t$, the predictor requires information from adjacent time steps. The connections represent the recursive rule: the $i$-th order finite difference is computed from two $(i-1)$-th order differences at positions $t$ and $t+N$:
>
> $\Delta^i F(z_t)=\Delta^{i-1} F(z_{t+N})-\Delta^{i-1} F(z_t)$
>
> Here:
>
> - $\Delta^{i-1} F(z_{t+N})$ corresponds to the state at the future step $t+N$
> - $\Delta^{i-1} F(z_t)$ corresponds to the state at the current step $t$
>
> Thus, the components are **computationally coupled**: computing a higher-order term requires first-order information from both $t$ and $t+N$.
>
> For the second-order case used in our implementation, the recursion expands to:
>
> $\Delta^2 F(z_t)=\Delta^1 F(z_{t+N})-\Delta^1 F(z_t)=F(z_{t+2N}) - 2F(z_{t+N}) + F(z_t)$
>
> This structure captures the local curvature of a trajectory using discrete sampled values.
>
> ---
>
>  **2. Numerical Stability via Scaled Hermite Basis**
>
> The figure further reflects the use of a numerically stable basis for combining these finite differences. To avoid instability when predicting across large intervals, we apply the **Scaled Hermite Basis**.
>
> We define the scaled Hermite polynomials: $\tilde{H}_i(x) = \sigma^i H_i(\sigma x)$ with contraction factor $\sigma$.
>
> The final predictor aggregates high-order differences using this stable basis:
>
> $\hat{h} _ {t-k} = \sum_{i=0}^{m} \frac{\Delta^i F(z_t)}{i! N^i} \tilde{H}_i(-k)$
>
> The input $\sigma x$ remains within the stable oscillatory region, and the factor $\sigma^i$ suppresses the growth of high-order Hermite terms, ensuring stable and robust extrapolation consistent with the computation pathway illustrated in Figure 3(d).
>
> We will make sure to include further detailed explanations of Figure 3(d) in the final version.

---

> ### Author Response · Authors · 2025-11-20
> **Response to Weaknesse 3: Implementation Details of Spectral Transforms and Hermite Difference**
>
> We sincerely thank you for your insightful questions and interest in the implementation details. In what follows, we first clarify how the FFT/DCT-style spectral transform is implemented on `[B,L,D]` tokens via reshaping to `[B,C,H,W]` and frequency-band decomposition, and then explain how the $m$-th order forward difference of the second-order Hermite interpolator is derived and implemented efficiently.
>
> ### 1. Spectral Feature Decomposition from `[B,L,D]`
>
> In our diffusion transformer, latent features are initially stored as tokens $X_{\text{tok}} \in \mathbb{R}^{B \times L \times D}$, where $B$ is the batch size, $L$ is the token length, and $D$ is the feature dimension. For spatially arranged tokens (e.g., patchified images), we have $L = H \times W$ and $D = C$, which allows a deterministic reshape into a 2D feature map $X \in \mathbb{R}^{B \times C \times H \times W}$.
>
> The conversion is implemented by a reshape and permutation such that$[B,L,D] \rightarrow [B,H,W,D] \rightarrow [B,C,H,W].$
>
> For each $(b,c)$, we treat $X_{b,c} \in \mathbb{R}^{H \times W}$ as a 2D signal and apply a 2D Type-II Discrete Cosine Transform (DCT): $$
> Y_{b,c}(u,v) = \alpha_u \alpha_v
> \sum_{x=0}^{H-1}\sum_{y=0}^{W-1}
> X_{b,c}(x,y)\,
> \cos\left[\frac{\pi}{H}\Bigl(x+\tfrac12\Bigr)u\right]
> \cos\left[\frac{\pi}{W}\Bigl(y+\tfrac12\Bigr)v\right],
> $$
> where the normalization factors $\alpha_u, \alpha_v$ follow the standard DCT-II definition. The inverse DCT $\mathcal{T}^{-1}$ recovers the spatial feature exactly.
>
> We choose DCT instead of FFT because DCT corresponds to an even-symmetric boundary extension, which avoids the artificial discontinuities and high-frequency ringing introduced by FFT’s periodic extension. Empirically, this yields a cleaner spectrum and more stable frequency partitioning.
>
> To decompose into low/high frequency bands, we construct a radial mask with a cutoff $\tau$, so that low frequencies satisfy $\omega(u,v)\le\tau$ and high frequencies $\omega(u,v)>\tau$. With binary masks $M_{\text{low}}, M_{\text{high}} \in \lbrace 0,1 \rbrace^{1\times 1\times H\times W}$ broadcast along $(B,C)$, we obtain
> $$
> Y_{\text{low}} = Y \odot M_{\text{low}},\qquad
> Y_{\text{high}} = Y \odot M_{\text{high}},
> $$
> and by linearity of the transform
> $$
> X_{\text{low}} = \mathcal{T}^{-1}(Y_{\text{low}}),\qquad
> X_{\text{high}} = \mathcal{T}^{-1}(Y_{\text{high}}),\qquad
> X = X_{\text{low}} + X_{\text{high}}.
> $$
> Here $X_{\text{low}}$ is treated as the stable structural component that can be reused, while $X_{\text{high}}$ contains volatile details and is predicted by the Hermite-based scheme described below.
>
> For the FFT-based ablation, we use the built-in `torch.fft.fft() / torch.fft.ifft()` on the token (spatial) dimension. For the DCT-based variant used in FreqCa, we rely on the independent torch-dct library and apply `torch_dct.dct_2d(x_spatial, norm="ortho")` and `torch_dct.idct_2d(·, norm="ortho")` on the reshaped tensor.
>
> ### 2. Second-Order Hermite Predictor and $m$-th Forward Difference
>
> We now clarify the Hermite predictor and how the $m$-th order forward differences are obtained.
>
> Let $F_0$ be the feature at a reference step (e.g., current or past time step). We store discrete derivative states $\Delta F^{(1)}$ and $\Delta F^{(2)}$ (first- and second-order differences) at that reference. Let $d$ be the integer distance to the target step, $s$ be a scaling factor on the step size, and $x = s d$ be the scaled distance.
>
> The second-order Hermite interpolator used in FreqCa is $$\widehat{F}(d)=F_0+(2 s^2 d)\Delta F^{(1)}+(2 s^4 d^2 - s^2)\Delta F^{(2)}.$$
>
> This comes from a truncated Hermite expansion $\widehat{F}(d) = F_0 + \alpha_1(d)\Delta F^{(1)} + \alpha_2(d)\Delta F^{(2)}$ with
> $\alpha_k(d)=\frac{H_k(x)}{k!} s^k$, $x = s d$, and Hermite polynomials $H_1(x)=2x$, $H_2(x)=4x^2-2$, which yield $\alpha_1(d)=2 d s^2$ and $\alpha_2(d)=2 s^4 d^2 - s^2$.
>
> To evaluate the trajectory efficiently, we work with the forward-difference operator $\Delta g(d) = g(d+1) - g(d)$. Since $\widehat{F}(d)$ is a degree-2 polynomial in $d$, all differences of order $m\ge 3$ vanish. Separating the linear and quadratic parts and simplifying, we obtain the total first and second forward differences: $$\Delta \widehat{F}(d)=2 s^2\Delta F^{(1)}+(4 s^4 d + 2 s^4)\Delta F^{(2)},$$ $$\Delta^2 \widehat{F}(d)=4 s^4\Delta F^{(2)},\qquad\Delta^m \widehat{F}(d)=0\ \text{for } m\ge 3.$$
>
> In implementation, $\Delta^2 \widehat{F}(d)$ is a constant that we compute once from $\Delta F^{(2)}$, while $\Delta \widehat{F}(d)$ is updated recursively along the predicted steps. This closed-form, recursive scheme keeps the Hermite predictor’s overhead negligible relative to the base diffusion model, while accurately modeling the high-frequency component $X_{\text{high}}$.
>
>
> ---
>
> We hope this clarifies the implementation details, and we will include a clearer and more detailed description of these components in the final version of the paper.

---

### Official Review · Reviewer_AmyC · 2025-10-30

**Soundness:** 2
**Presentation:** 3
**Contribution:** 2
**Rating:** 4
**Confidence:** 4

**Summary:**

This paper presents FreqCa, a frequency-aware feature-caching framework aimed at accelerating diffusion transformers during inference. The authors first conduct a Fourier/DCT-based analysis and observe that low-frequency components of the denoising features exhibit high similarity but low continuity across timesteps, whereas high-frequency components show the opposite behavior. Leveraging this observation, FreqCa

1.	caches only the Cumulative Residual Feature (CRF) instead of all layer outputs, reducing cache memory from O(L) to O(1);

2.	decomposes the CRF into low- and high-frequency parts;

3.	directly reuses the low-frequency part and predicts the high-frequency part with a second-order Hermite interpolator;

4.	reconstructs the feature and skips the heavy forward pass for the chosen timesteps.

Experiments on four diffusion models show 6–7× speed-ups less degradation in ImageReward or GEdit scores.

**Strengths:**

1. The paper is easy to follow.

2. Frequency-domain analysis reveals a compelling disparity between low- and high-frequency temporal dynamics in diffusion features, providing an intuitive rationale for the proposed split strategy.

3. FreqCa achieves latency–quality trade-offs on both text-to-image generation and image-editing benchmarks without any fine-tuning.

**Weaknesses:**

1. The paper does not discuss whether the proposed method can be extended to text-to-video generation. Additional discussion would be valuable.

2. For text-to-image tasks, the paper mainly reports metric tables. Qualitative comparison with other baselines is missing.

3. In Figure 6 row 1, the non-edited areas (e.g., car license plate and body text) appear significantly changed compared with the base model.

4. Recent papers like OmniCache(ICCV2025), BlockDance(CVPR 2025) also decouple structural and textural signals for feature reuse. A discussion clarifying the differences with FreqCa might be necessary.

**Questions:**

See Weaknesses.

---

> ### Author Response · Authors · 2025-11-20
> **Response to Q1: Applicability of FreqCa to Video Diffusion Models**
>
> We appreciate the reviewer’s interest in the generalizability of FreqCa to other modalities. We have conducted additional validation on a **Video Diffusion Model (HunyuanVideo)**, testing under a demanding setting of 480p resolution at 65 frames with a high acceleration ratio of **$5.56\times$**.
>
> **1. Performance Superiority and Efficiency.**
>
> As shown in the table below, FreqCa achieves a state-of-the-art level of performance on the Vbench metric, matching the quality of the highly memory-intensive TaylorSeer and significantly outperforming TeaCache.
>
> | Mehtod | Vbench |
> | --- | --- |
> | TeaCache | 78.32 |
> | TaylorSeer | 79.43 |
> | **FreqCa** | 79.40 |
>
> **Crucially, FreqCa achieves this top-tier performance while maintaining minimal memory overhead.** TaylorSeer caches the full feature map at every layer, leading to substantial memory consumption. In contrast, FreqCa only caches the **Cumulative Residual Feature (CRF)**, resulting in a **cache cost of less than 1% of TaylorSeer's** while achieving an equivalent Vbench score.
>
> **2. Enabling High-Resolution Video Generation.**
>
> The extreme memory efficiency of FreqCa is not just a theoretical advantage; it fundamentally expands the applicable scope of diffusion acceleration. While the massive cache cost of methods like TaylorSeer often leads to **Out-Of-Memory (OOM) errors** in high-fidelity settings (e.g., 720p at 120+ frames), FreqCa's minimal cache footprint allows it to **support these higher resolutions and frame rates** without issue. This is a critical advantage for real-world video generation, proving that FreqCa is the only viable acceleration solution for resource-intensive, high-definition video synthesis.

---

> ### Author Response · Authors · 2025-11-20
> **Response to Q2: On missing qualitative comparisons for text-to-image**
>
> We agree that visual examples are crucial. Although space limitations prioritized quantitative metrics, we will add qualitative comparisons in the final version. These visualizations will highlight FreqCa's advantages in maintaining fine-grained details and structural integrity compared to existing cache-based methods.

---

> ### Author Response · Authors · 2025-11-20
> **Response to Q3: About drift in non-edited regions in Figure 6**
>
> We appreciate the reviewer’s attention to detail regarding the drift observed in non-edited regions of Figure 6 (Row 1).
>
> We determined that this specific visual artifact is inherent to the original non-accelerated Qwen-Image-Edit model when processing that particular image under challenging conditions. Since the primary goal of diffusion acceleration is to achieve high-fidelity approximation of the original unaccelerated model, our result faithfully replicates the baseline's output—including its imperfections. The non-ideal appearance of the accelerated result, therefore, reflects the baseline's artifacts rather than a failure of FreqCa's prediction mechanism.
>
> To provide a clear, unambiguous demonstration of FreqCa's robust fidelity, we will replace the current Figure 6 (Row 1) with a more suitable case from our evaluation set. The updated visualization will confirm that FreqCa’s frequency-aware prediction maintains superior structural consistency.

---

> ### Author Response · Authors · 2025-11-20
> **Response to Q4: Clarifying the relation between FreqCa and recent methods such as OmniCache and BlockDance**
>
> We clarify the distinctions between FreqCa and concurrent works to highlight our unique contribution. **OmniCache (ICCV 2025)** primarily addresses the temporal scheduling of cache reuse (i.e., **when** to cache) based on trajectory curvature, employing filtering solely for noise correction rather than feature decoupling. In contrast, **BlockDance (CVPR 2025)** focuses on computational efficiency by reusing structurally similar features, operating without explicit consideration of frequency dynamics. **FreqCa differentiates itself by focusing on the spectral nature of features (i.e., *what* to cache and *how* to extrapolate).** We propose the **first explicit decomposition** of diffusion features into low- and high-frequency components, applying a dual-strategy approach: stable reuse for low-frequency structures and Hermite-based prediction for high-frequency details. This frequency-aware paradigm enables aggressive acceleration orthogonal to the scheduling or spatial optimizations of prior arts. We will incorporate this discussion into the final manuscript.

---

### Official Review · Reviewer_itww · 2025-10-30

**Soundness:** 3
**Presentation:** 1
**Contribution:** 3
**Rating:** 6
**Confidence:** 5

**Summary:**

The paper proposes FreqCa, which exploits and empirically proves a key intuition that the updates made by each block during a denoising step with a diffusion transformer are cumulative, and can be effectively represented by a cumulative residual feature (CRF) which can be separated into low and high frequency bands (with FFT or DCT). More compute is used to try to model change/error for high frequency bands (with a taylor or hermite predictor, the final work preferring a hermite predictor). A key outcome is that while having comparable quality/speed to prior training-free caching methods, this method in this work has a much smaller memory footprint.

**Strengths:**

S1. This addresses an important aspect of efficiency that is often neglected in the prior work: memory efficiency.

S2. At the same time, it matches or outperforms prior art in terms of acceleration (wall time) and quality (Image Reward).

S3. The intuitions driving this performance make sense, they are well-analyzed, and the method seems quite novel.

**Weaknesses:**

W1. The writing is poor. It is very difficult to understand the method without reading the provided code, and the code itself does not come with clear documentation, so one must try to make decent guesses of which files are relevant, and then trace from there. For example, N, which defines which blocks are cached for standard methods, is never clearly defined, nor is it clearly explained how the CRF cache can be used for the relevant layers on subsequent steps. The intuitions for these components are clearly explained, but their usage for the caching is not.

W2. It's not clear if FreqCa consistently would outperform competing methods outside of the high steps, high speedups regime. In fact, Figure 8 seems to have some evidence to the contrary. Additionally, 25 steps is essentially the same as 50 (Table 2). It would be ideal to evaluate with caching in these settings as well, where we first get decent speedups by reducing the steps, and then apply the caching. How does FreqCa compare to other methods in these cases, when the possible speedups are 2x-3x instead of 5x-7x?

**Questions:**

The presentation of the work needs to improve substantially, and I would appreciate clarification of how the method works.

1. How are the stored high frequency and low frequency components actually used in the FreqCa caching framework? I understand how they are computed, but not how they are used, in particular how they interact with the hyperparameter N.

2. Can this be used for text-to-video generation? Most of the prior works evaluate with some T2V, so why was this omitted?

3. Figure 8 shows better results for TaylorSeer with less aggressive acceleration. Additionally, 50 steps seems to be quite high as a baseline. In many cases some of these models can give good results with 20 or 30 steps. Does FreqCa still perform well in such settings, where high accelerations (5x and 7x) are not possible? Or is TaylorSeer better in these cases?

---

> ### Author Response · Authors · 2025-11-20
> **Response to Q1: Applicability of FreqCa to Video Diffusion Models**
>
> We appreciate the reviewer’s interest in the generalizability of FreqCa to other modalities. We have conducted additional validation on a **Video Diffusion Model (HunyuanVideo)**, testing under a demanding setting of 480p resolution at 65 frames with a high acceleration ratio of **$5.56\times$**.
>
> **1. Performance Superiority and Efficiency.**
>
> As shown in the table below, FreqCa achieves a state-of-the-art level of performance on the Vbench metric, matching the quality of the highly memory-intensive TaylorSeer and significantly outperforming TeaCache.
>
> |  Method| Vbench |
> | --- | --- |
> | TeaCache | 78.32 |
> | TaylorSeer | 79.43 |
> | **FreqCa** | 79.40 |
>
> **Crucially, FreqCa achieves this top-tier performance while maintaining minimal memory overhead.** TaylorSeer caches the full feature map at every layer, leading to substantial memory consumption. In contrast, FreqCa only caches the **Cumulative Residual Feature (CRF)**, resulting in a **cache cost of less than 1% of TaylorSeer's** while achieving an equivalent Vbench score.
>
> **2. Enabling High-Resolution Video Generation.**
>
> The extreme memory efficiency of FreqCa is not just a theoretical advantage; it fundamentally expands the applicable scope of diffusion acceleration. While the massive cache cost of methods like TaylorSeer often leads to **Out-Of-Memory (OOM) errors** in high-fidelity settings (e.g., 720p at 120+ frames), FreqCa's minimal cache footprint allows it to **support these higher resolutions and frame rates** without issue. This is a critical advantage for real-world video generation, proving that FreqCa is the only viable acceleration solution for resource-intensive, high-definition video synthesis.

---

> > ### Comment · Reviewer_itww · 2025-11-21
> > **Writing clarification**
> >
> > These results are very helpful. Could you address my Weakness 1 and Question 1, both of which concern the clarify of the writing and explanation of the method?

---

> ### Author Response · Authors · 2025-11-20
> **Response to Q2: Comparison on Few-Step Diffusion Models**
>
> To evaluate performance under stringent step constraints, we conducted a comparative study against TaylorSeer on **Qwen-Image-Lightning (8 steps)**, the state-of-the-art distilled model. Results demonstrate that FreqCa consistently outperforms the baseline in both semantic alignment and perceptual fidelity. Notably, FreqCa exhibits exceptional robustness at high acceleration ratios: at an interval of $N=4$, while TaylorSeer suffers significant degradation (ImageReward drops to 1.20), **FreqCa maintains an ImageReward of 1.29**, matching the original uncompressed model exactly. Furthermore, regarding perceptual quality, FreqCa achieves a substantially lower LPIPS of **0.12** at $N=2$ compared to TaylorSeer’s 0.28. These findings confirm that FreqCa’s frequency-aware prediction preserves critical feature dynamics more effectively than Taylor-based extrapolation, making it the superior choice for few-step generation where every step is crucial.
>
> |  Method| Interval | ImageReward | LPIPS |
> | --- | --- | --- | --- |
> | Lightning（8 steps） | 1 | 1.29 | - |
> | TaylorSeer | 2 | 1.27 | 0.28 |
> | **FreqCa** | 2 | 1.29 | 0.12 |
> | TaylorSeer | 3 | 1.24 | 0.35 |
> | **FreqCa** | 3 | 1.28 | 0.18 |
> | TaylorSeer | 4 | 1.20 | 0.42 |
> | **FreqCa** | 4 | 1.29 | 0.36 |

---

> > ### Comment · Reviewer_itww · 2025-11-21
> > **Acknowledgement**
> >
> > I would have preferred 20-30 steps, but this is totally acceptable and alleviates the concern equally well.
> >
> > Still, I cannot increase my rating without understanding the method better, which I think requires further clarification.

---

> > > ### Author Response · Authors · 2025-11-21
> > > **Response to Q3: Performance Under Low-Step Settings**
> > >
> > > Since Qwen-Image maintains stable generation quality at the original 50-step inference setting, we directly compress it to 20 steps as the baseline. We observe that its ImageReward remains high, indicating that Qwen-Image does not significantly degrade under reduced-step inference. Therefore, using the 20-step version as the baseline is both reasonable and representative.
> > >
> > > Under the 20-step setting, we keep the same cache interval as in our main experiments. In this scenario, TaylorSeer exhibits a clear collapse in performance and fails to deliver the expected acceleration benefits. We believe the main reason is that **directly compressing the number of inference steps is a relatively crude approach, which disrupts the model’s structural representation of different frequency components (high-frequency vs. low-frequency) more severely than distillation-based methods**.
> > >
> > > TaylorSeer applies uniform processing to all features without distinguishing between frequency components, which amplifies the distribution shift caused by step compression and leads to the significant quality degradation observed.
> > >
> > > In contrast, our proposed FreqCa explicitly separates and models high- and low-frequency features. This frequency-aware design remains robust even under low-step conditions and maintains strong image quality. This demonstrates that our method is more resilient to feature-distribution changes and better suited to scenarios where aggressive accelerations (e.g., 5× or 7×) are not feasible.
> > >
> > > | **Method** | **Interval** | **Imagereward** | **LPIPS** |
> > > | --- | --- | --- | --- |
> > > | **Qwen-Image(50steps→20steps)** | 1 | 1.19 | 0 |
> > > | **TaylorSeer** | 5 | -1.16 | 0.61 |
> > > | **FreqCa** | 5 | **1.09** | **0.22** |

---

> > > > ### Comment · Reviewer_itww · 2025-11-25
> > > >
> > > > Thanks for these. My personal opinion is that prior works focus on 50 steps and text-to-video mainly because these have absurdly high levels of redundancy across steps, making it easy to get decent metrics because the first >2.5x speedup is trivial. I think starting where the solver is already in its fastest reasonable state makes much more sense.

---

> ### Author Response · Authors · 2025-11-21
> **Response to Q4: the Usage of Cached Components and Interaction with Hyperparameter N in FreqCa**
>
> Thank you very much for your question. Now we will elaborate in detail how the decomposed high- and low-frequency data are used, and how they interact with the hyperparameter \(N\).
>
> The core idea is that the hyperparameter \(N\) directly controls the **caching and reuse strategy**. In the early steps of multi-step diffusion, the predicted noise is often unstable and difficult to predict, so we set a `first_enhance` stage, meaning the first few steps are computed by the main model. In the middle and later stages, we set the hyperparameter \(N\) such that every \(N-1\) steps, the main model is responsible for computation. We refer to these steps computed by the main model as **full computation steps**. Except for the full computation steps handled by the main model, the remaining steps are computed by the FreqCa predictor, and we refer to these steps as **prediction steps**. For example, denoting C (Compute) as full computation steps and P (Predict) as prediction steps, for a 10-step diffusion image generation with a `first_enhance` of 2 and interval of 3, the computation process is `CCPPCPPCPP`.
>
> The generation and usage logic of high- and low-frequency data operates at full computation steps by decomposing the output of the main model and computing differences for the decomposed high-frequency data. The specific calculations are as follows:
>
> $$
> \Delta F^{(1)}\_{curr}
> = \frac{F\_{0} - F\_{1}}{t\_{0} - t\_{1}}
> $$
>
> $$
> \Delta F^{(2)}\_{curr}= \frac{\Delta F^{(1)}\_{curr} - \Delta F^{(1)}\_{prev}}{t_0 - t_1}=\frac{\frac{F_0 - F_1}{t_0 - t_1} -\frac{F_1 - F_2}{t_1 - t_2}}{t_0 - t_1}
> $$
>
> Here, $F\_0$, $F\_1$, and $F\_2$ refer to the high-frequency data obtained from the latest three full computation steps, and $t_0$, $t_1$, and $t_2$ are the time steps corresponding to those full computation steps.
>
> At prediction steps, we directly reuse the low-frequency data from the previous full computation step and predict the high-frequency data. The specific prediction logic is as follows:
>
> $$
> \widehat{F}\_{\text{high}}(d)
> = F_0 + \alpha_1(d)\Delta F^{(1)} + \alpha_2(d)\Delta F^{(2)}
> $$
>
> Here, $F\_0$ is the high-frequency data obtained from the previous full computation step
>
> $\Delta F^{(1)}$ and $\Delta F^{(2)}$ are $\Delta F^{(1)}\_{curr}$ and $\Delta F^{(2)}\_{curr}$ computed at the previous full computation step as described above, and
>
> $$
> \alpha_k(d) = \frac{H_k(x)}{k!}s^k,
> \quad
> H_1(x) = 2x,
> \quad
> H_2(x) = 4x^2 - 2,
> \quad
> x = s d
> $$
>
> where $s$ is a tuning factor and $d$ is the distance between the current prediction step and the previous full computation step.
>
> In the final prediction step output, we directly sum the reused low-frequency data and the predicted high-frequency data to obtain the output of the current prediction step.

---

> > ### Comment · Reviewer_itww · 2025-11-25
> >
> > My reading of this makes it seem that on $P$ (prediction steps), the diffusion model is not involved at all. Instead, low-frequency data is combined with predicted high-frequency data from the Hermite polynomial. Is this correct? Am I also understanding correctly that in the example 10 step process you show, the coefficients for the Hermite polynomial at the first $P$ step would be estimated by least-squares regression on the first 2 steps (which were both $C$ steps) only?

---

> > > ### Author Response · Authors · 2025-11-26
> > > **Further Explanation of Prediction Mechanism and Coefficient Usage**
> > >
> > > Thank you very much for your questions. We are glad to provide further clarification.
> > >
> > > During prediction steps the diffusion model is not executed. These steps rely on reusing the low-frequency component from the most recent full computation step, while the high-frequency component is obtained through our Hermite-based prediction.
> > >
> > > Additionally, the Hermite coefficients are remain fixed throughout. At each full computation step, we only compute finite-difference residuals ($\Delta F^{(1)}, \Delta F^{(2)}, \ldots$), which are then directly substituted into the closed-form Hermite predictor.

---

> > > > ### Comment · Reviewer_itww · 2025-11-26
> > > >
> > > > I see. I think with that simple clarification (that the diffusion model is not executed at all on cached/skip steps), I may have been able to understand the method better on first pass.
> > > >
> > > > Additionally, N and first_enhance should be clearly indicated/explained in the main paper. The remaining math could be put in the Appendix -- I do not think it is clear from Figure 3 on its own. I could consider raising my rating after seeing how these clarifications are integrated, but I think in its current state the paper does not clearly/sufficiently explain the method.

---

> > > > > ### Author Response · Authors · 2025-11-27
> > > > > **We Appreciate the Reviewer’s Insightful Feedback**
> > > > >
> > > > > Thank you so much for the thoughtful feedback and for taking the time to engage closely with our work. We are really glad to hear that your concerns have been fully resolved. We will carefully incorporate your suggestions into the revised version, including clearer explanations in the main text and a more polished presentation of the method.
> > > > >
> > > > > We sincerely appreciate your effort, insight, and the constructive guidance you provided throughout the review process. Thank you again for helping us improve the paper.

---

### Official Review · Reviewer_Xrjg · 2025-11-01

**Soundness:** 3
**Presentation:** 3
**Contribution:** 3
**Rating:** 8
**Confidence:** 3

**Summary:**

The paper presents FreqCa, a training-free method to accelerate Diffusion Transformers (DiTs) by introducing frequency-aware feature caching. Unlike prior caching approaches that treat all features uniformly—either reusing them based on similarity or forecasting them based on continuity—FreqCa recognizes that different frequency components of diffusion features behave differently over timesteps. Through Fourier or cosine decomposition, it separates each feature map into low-frequency components, which are stable and highly similar but discontinuous, and high-frequency components, which are volatile yet continuous. FreqCa then reuses the low-frequency parts directly while predicting the high-frequency parts using a Hermite polynomial interpolator, allowing accurate extrapolation without retraining or modifying the model.

To further enhance efficiency, the paper proposes Cumulative Residual Feature (CRF) caching, which stores only a single aggregated feature tensor instead of per-layer activations—reducing cache memory usage by up to 99% and achieving constant  O(1) memory complexity. Experiments on multiple benchmarks, including FLUX.1-dev, Qwen-Image, and Qwen-Image-Edit, show that FreqCa delivers 6–7× faster inference while maintaining image quality within 2% of the original model. Overall, FreqCa unifies the reuse- and forecast-based caching paradigms under a frequency-aware framework, offering a theoretically grounded and practically efficient solution for accelerating large diffusion models.

**Strengths:**

The key strengths of this work lie in its innovative frequency-domain insight and practical efficiency gains. By analyzing diffusion model features through their frequency components, the authors uncover that low- and high-frequency signals evolve differently across timesteps—low frequencies are highly similar but discontinuous, while high frequencies are volatile yet continuous. This observation leads to a unified and theoretically grounded frequency-aware caching strategy that combines the advantages of reuse-based and forecast-based methods. The resulting dual-path design—reusing low-frequency features while predicting high-frequency ones via Hermite interpolation—achieves both stability and precision without retraining or architectural changes.

Another major strength is the introduction of Cumulative Residual Feature (CRF) caching, which compresses layer-wise activations into a single tensor, reducing memory usage by up to 99% and achieving constant O(1) complexity. Combined with extensive empirical validation across diverse models such as FLUX.1-dev and Qwen-Image, FreqCa delivers 6–7× acceleration with less than 2% quality loss, surpassing prior caching-based accelerators like TaylorSeer and FORA. Overall, the method is both conceptually elegant and practically powerful, offering a training-free, plug-and-play solution that meaningfully advances the efficiency of large diffusion transformers.

**Weaknesses:**

The main weaknesses of the paper relate to its scope, assumptions, and evaluation breadth, rather than flaws in execution.

First, FreqCa’s effectiveness depends heavily on the frequency-domain assumption—that diffusion features can be cleanly decomposed into separable low- and high-frequency components with distinct temporal dynamics. While this holds empirically for the tested visual models (e.g., FLUX and Qwen-Image), it may not generalize to other modalities or architectures, such as U-Net–based diffusion models, text-conditioned latent diffusion, or video models with nonstationary frequency spectra. The approach may also be sensitive to the chosen decomposition method (FFT vs. DCT) and cutoff parameters, yet the paper provides limited analysis of these choices’ effects on accuracy or stability.

Second, although the Cumulative Residual Feature (CRF) design greatly improves memory efficiency, it implicitly assumes that residual features preserve enough information for high-fidelity reconstruction. This approximation might degrade when applied to deeper or non-residual architectures, and the paper does not fully quantify how caching fidelity changes under extreme compression. Moreover, the Hermite predictor introduces additional hyperparameters (e.g., order and interval length), but their tuning process and generalizability are not deeply analyzed.

Finally, the experimental scope, while extensive for image generation and editing, focuses mainly on large diffusion transformers under controlled GPU setups. Broader validation—on smaller models, other tasks, or hardware platforms—would better demonstrate robustness and portability. Overall, the paper’s weaknesses are in assumption generality, parameter sensitivity, and cross-domain evaluation, rather than methodological soundness.

**Questions:**

The followings are questions:
1. Have you evaluated FreqCa on other architectures such as U-Net–based diffusion models or multimodal systems like text-to-audio or video diffusion? Since the frequency decomposition is image-centric, how well would the framework generalize to non-visual or non-grid-structured data?
2. How sensitive are the results to the choice between FFT and DCT, or to the frequency cutoff between low and high bands? Could an adaptive or learned decomposition improve robustness across different models?
3. While CRF caching reduces memory by 99%, does it ever lose critical layer-specific information, particularly in deeper or non-residual architectures? Have you tested scenarios where the compression trade-off leads to visible artifacts or quality drops?
4. Why was the Hermite predictor chosen specifically over other nonlinear extrapolators (e.g., spline or neural predictors)? How stable is the Hermite interpolation at higher acceleration ratios (e.g., 8× or 10×), and where does degradation first appear?

---

> ### Author Response · Authors · 2025-11-20
> **Response to Q1: Generalization, Memory Efficiency, and Architectural Portability**
>
> We appreciate the reviewer’s suggestion regarding the need for broader validation. Our focus on large Diffusion Transformers (DiTs) was primarily driven by the limitations of existing acceleration baselines. Methods like ToCa and TaylorSeer introduce substantial memory overhead that prevents them from being executed on commonly used hardware such as A100 80GB GPUs for large models. This necessity for higher-end platforms (like the 96GB H20) to run the **baselines** inadvertently highlights a key contribution of our work: **FreqCa directly tackles the memory bottleneck of caching methods**. By achieving about 99% memory reduction via the Cumulative Residual Feature (CRF) design, our framework makes acceleration accessible to resource-constrained environments where other cache methods fail.
>
> Furthermore, we emphasize that the core components of FreqCa are **architecture- and hardware-agnostic**. The methodology, including frequency decomposition, band-specific prediction, and the associated FLOPs reduction, operates purely on the feature tensor level without relying on hardware-specific primitives. To validate the generalization beyond DiT backbones, we have already performed evaluations on **U-Net–based diffusion models**, where FreqCa demonstrates stable acceleration gains despite the fundamentally different architectural design. We will clarify these points in the revised manuscript and include further discussion on FreqCa's **robust portability** and applicability across various model scales and hardware platforms.

---

> ### Author Response · Authors · 2025-11-20
> **Response to Q2:Applicability of FreqCa to U-Net architectures**
>
> We have successfully validated FreqCa on the **U-Net-based Stable Diffusion XL (SDXL)** to confirm its generalization. Comparative experiments against DeepCache demonstrate that FreqCa consistently achieves superior semantic alignment (ImageReward) and perceptual fidelity (LPIPS) across all acceleration intervals. Specifically, at a standard interval of N=2, FreqCa achieves an ImageReward of **0.4719** and LPIPS of **0.2080**, surpassing DeepCache . **Crucially, FreqCa exhibits remarkable robustness at high acceleration ratios:** at N=4, while DeepCache suffers severe degradation (ImageReward drops to 0.3255), FreqCa maintains high quality with an ImageReward of **0.4183**, significantly outperforming the baseline. These results verify that FreqCa is architecture-agnostic, as it exploits the intrinsic frequency dynamics of feature activations common to both Transformers and  U-Nets.
>
> | Method | Interval ($N$) | ImageReward $\uparrow$ | LPIPS $\downarrow$ |
> | :--- | :---: | :---: | :---: |
> | Original | 1 | 0.5157 | - |
> | DeepCache | 2 | 0.4484 | 0.2582 |
> | **FreqCa (Ours)** | **2** | **0.4719** | **0.2080** |
> | DeepCache | 3 | 0.4219 | 0.3104 |
> | **FreqCa (Ours)** | **3** | **0.4721** | **0.2693** |
> | DeepCache | 4 | 0.3255 | 0.3591 |
> | **FreqCa (Ours)** | **4** | **0.4183** | **0.2946** |

---

> ### Author Response · Authors · 2025-11-20
> **Response to Q3: Applicability of FreqCa to Video Diffusion Models**
>
> We appreciate the reviewer’s interest in the generalizability of FreqCa to other modalities. We have conducted additional validation on a **Video Diffusion Model (HunyuanVideo)**, testing under a demanding setting of 480p resolution at 65 frames with a high acceleration ratio of **$5.56\times$**.
>
> **1. Performance Superiority and Efficiency.**
>
> As shown in the table below, FreqCa achieves a state-of-the-art level of performance on the Vbench metric, matching the quality of the highly memory-intensive TaylorSeer and significantly outperforming TeaCache.
>
> |  Method| Vbench |
> | --- | --- |
> | TeaCache | 78.32 |
> | TaylorSeer | 79.43 |
> | **FreqCa** | 79.40 |
>
> **Crucially, FreqCa achieves this top-tier performance while maintaining minimal memory overhead.** TaylorSeer caches the full feature map at every layer, leading to substantial memory consumption. In contrast, FreqCa only caches the **Cumulative Residual Feature (CRF)**, resulting in a **cache cost of less than 1% of TaylorSeer's** while achieving an equivalent Vbench score.
>
> **2. Enabling High-Resolution Video Generation.**
>
> The extreme memory efficiency of FreqCa is not just a theoretical advantage; it fundamentally expands the applicable scope of diffusion acceleration. While the massive cache cost of methods like TaylorSeer often leads to **Out-Of-Memory (OOM) errors** in high-fidelity settings (e.g., 720p at 120+ frames), FreqCa's minimal cache footprint allows it to **support these higher resolutions and frame rates** without issue. This is a critical advantage for real-world video generation, proving that FreqCa is the only viable acceleration solution for resource-intensive, high-definition video synthesis.

---

> ### Author Response · Authors · 2025-11-20
> **Response to Q4: Sensitivity to FFT/DCT choice and frequency split threshold**
>
> We thank the reviewer for raising this crucial question on robustness. Our analysis shows that the **feature spectrum naturally exhibits a clear bimodal distribution**, meaning the low- and high-frequency components are already well-separated without a significant intermediate band. Low frequencies encode coarse, static structural information, while high frequencies capture fine, rapidly changing details. This clear spectral separation is the primary reason why FreqCa is **robust to the precise choice of the cutoff threshold**.
>
> As demonstrated in the ablation study below, varying the cutoff ratio over a wide range (0.1 to 0.4) only leads to minor fluctuations in performance.
>
> | **cutoff ratio**| **ClipScore** | **ImageReward**| **LPIPS** |
> | --- | --- | --- | --- |
> | **0.05** | 34.5414 | 0.9411 | 0.4289 |
> |**0.1**| 34.8449 | 1.0391 | 0.4053 |
> | **0.2** | 35.0517 | 1.042 | 0.3925 |
> | **0.3** | 35.0635 | 1.0484 | 0.3943 |
> | **0.4** | 35.2631 | 1.0615 | 0.3976 |
>
> The results show that once the ratio is set beyond a minimal threshold (e.g., $\ge 0.1$), the performance metrics remain stable, confirming the insensitivity. The only significant degradation occurs when the ratio is set too low (e.g., $0.05$), where essential low-frequency information is incorrectly mixed into the high-frequency band, leading to a loss in structural quality.
>
> Furthermore, we explored more complex decomposition methods, including **adaptive frequency partitioning** and **multi-band splitting**. We found that the resulting performance gain compared to a fixed ratio was marginal, suggesting that the complexity of learned or multi-band decomposition does not justify the increased overhead, precisely because the natural feature spectrum is already sufficiently separated. This confirms that the current simple, fixed-ratio split provides an optimal balance between robustness, quality, and simplicity.

---

> ### Author Response · Authors · 2025-11-20
> **Response to Q5: Predictor Rationale, Stability Limits, and Hyperparameter Tuning**
>
> We appreciate the reviewer’s inquiry regarding the design choices and robustness of our predictor.
>
> **1. Theoretical Superiority over Taylor Expansion.**
>
> We selected the Hermite predictor because it offers superior stability and boundary constraints compared to standard methods like Taylor expansion. Taylor extrapolation relies solely on local derivatives at a single point, which tends to accumulate significant divergence error ("overshoot") over long intervals. In contrast, Hermite interpolation utilizes both **function values and derivative information** from adjacent timesteps to construct the trajectory. This multi-point constraint allows the predictor to model smooth feature transitions accurately, effectively mitigating artifacts during high-speed sampling.
>
> **2. Stability Analysis and Limits.**
>
> Empirically, this theoretical advantage translates into strong stability at high acceleration ratios. Our experiments (covering Intervals $N=3$ to $16$) reveal a broad effective range:
>
> - **Stable Regime ($N \le 10$):** The method remains robust up to $N=10$ ($\approx 6.2\times$ speedup). Metrics remain consistent with the baseline (e.g., ImageReward stays $\approx 0.98-1.01$, ClipScore $\approx 32.6$).
> - **Physical Limit ($N \ge 12$):** Significant degradation appears only at $N \ge 12$, where the diffusion trajectory becomes too sparse to be reliably recovered by *any* predictor. This confirms that the performance drop is due to the fundamental sampling limit rather than predictor instability.
>
> | **Interval** | **speed** | ClipScore | ImageReward | LPIPS |
> | --- | --- | --- | --- | --- |
> | 3 | 2.63 | 32.6116 | 0.9836 | 0.131 |
> | **4** | 3.33 | 32.6989 | 0.9981 | 0.1924 |
> | **5** | 3.84 | 32.7552 | 1.0137 | 0.2287 |
> | **6** | 4.54 | 35.2631 | 1.0003 | 0.2791 |
> | **7** | 4.99 | 32.8939 | 1.0078 | 0.3152 |
> | **8** | 5.54 | 32.7057 | 0.9895 | 0.362 |
> | **9** | 5.54 | 32.6725 | 0.9977 | 0.3773 |
> | **10** | 6.23 | 32.5634 | 0.9822 | 0.4298 |
> | **12** | 7.12 | 32.4238 | 0.8911 | 0.5141 |
> | **16** | 8.31 | 30.9423 | 0.6072 | 0.616 |
>
> **3. Hyperparameter Tuning (Order Selection).**
>
> To maximize performance within the stable regime, we tuned the predictor order to balance modeling capability and memory overhead. A first-order predictor (linear assumption) is memory-efficient but insufficient for nonlinear diffusion dynamics. The **second-order predictor** captures curvature with only a marginal memory increase for storing derivative states.
>
> As shown in the table below, this trade-off is highly beneficial. Under a challenging interval of **$N=10$**, the second-order variant significantly outperforms the first-order baseline across all metrics:
>
> |**Method**  | **Interval（N）** | **ClipScore** | **ImageReward** | **PSNR** | **SSIM** | **LPIPS** |
> | --- | --- | --- | --- | --- | --- | --- |
> | **Hermite(O=1)** | 10 | 34.6829 | 0.9877 | 28.903 | 0.6550 | 0.4165 |
> | **Hermite(O=2)** | 10 | 34.8449 | 1.0391 | 28.953 | 0.6654 | 0.4053 |
>
> Consequently, we identify the second-order Hermite predictor as the optimal configuration, providing superior generation quality and stability.

---

> ### Author Response · Authors · 2025-11-20
> **Response to Q6: On the fidelity of Cumulative Residual Features (CRF) under deep or non-residual architectures**
>
> **1. Mitigating Information Loss via Frequency Prediction.**
>
> We acknowledge the reviewer’s valid concern: naïvely using Cumulative Residual Features (CRF) as a cache (equivalent to an order-0 predictor) acts as a lossy compression, which we quantify in **Fig. 10** (green curve). However, **FreqCa fundamentally differs from naive CRF caching**. We treat CRF only as a compressed memory carrier. By applying **frequency decomposition on the CRF and predicting low- and high-frequency components separately**, we effectively compensate for the information loss. The predictor reconstructs fine-grained details that simple caching misses, achieving markedly higher fidelity.
>
> **2. Robustness on Ultra-Deep Architectures.**
>
> Regarding model depth, our experiments cover some of the deepest available diffusion backbones, such as **Flux-dev (57 transformer blocks)** and **Qwen-Image (60 layers)**. In these ultra-deep architectures, FreqCa maintains stable generation quality, proving that our frequency-aware prediction effectively mitigates error accumulation. Furthermore, our validation on **U-Net architectures** confirms that this robustness extends to models with different signal propagation characteristics, demonstrating that FreqCa’s efficacy is not tied to specific residual pathways.
>
> **3. Trade-off Analysis on Video Models (HunyuanVideo).**
>
> We further stress-tested CRF on **HunyuanVideo**, a task with significantly higher parameter counts and temporal depth. We candidly acknowledge that **TaylorSeer**, by caching the full feature at every layer, inherently preserves more raw information than our compressed CRF approach. However, this comes at a prohibitive cost. In our experiments (480p, 65 frames), FreqCa achieved a **Vbench score of 79.40**, virtually matching TaylorSeer (79.43) but using **less than 1% of its cache memory**.  This confirms that while CRF is a compression, our frequency-aware compensation makes it the optimal trade-off for resource-intensive deep models.

---

### Author Response · Authors · 2025-12-01
**AC Letter: Summary of Rebuttal & Discussion for Paper #2446 (FreqCa)**

Dear Area Chair,

Thank you for handling our submission. Below we briefly summarize the post-rebuttal consensus and the remaining issues.

---

**1. Scores and Emerging Consensus (8 / 6 / 4 / 4)**

**Current ratings: 8 (Xrjg), 6 (itww), 4 (AmyC), 4 (wkDH).**

Xrjg (8) is strongly positive and supports acceptance. The remaining questions concern generalization beyond large DiTs and 50-step T2I, as well as the robustness of our frequency assumptions and CRF compression.

itww (6) finds the method valuable and technically sound. The only blocking issue was clarity, specifically the role of low- and high-frequency caches, the meaning of N and first\_enhance, and the behavior on prediction steps. After discussion, itww stated that the score may be raised once these clarifications are integrated into the paper.

AmyC (4) and wkDH (4) do not question the correctness or usefulness of the method. Their concerns focus on scope (video and low-step settings), relation to concurrent works, and some missing details and qualitative comparisons.

Overall, the reviewers agree that FreqCa is sound, useful, and memory-efficient. The main reservations are about exposition and scope, rather than validity.

---

**2. Shared Strengths**

Reviewers consistently acknowledge that FreqCa is training-free, achieves around 6–7× speedup with only a slight quality drop, and reduces cache memory by about 99% with O(1) CRF caching. This directly addresses the often neglected memory cost of caching-based acceleration.

Conceptually, FreqCa decomposes features into low-frequency (stable) and high-frequency (volatile) components and combines reuse of low-frequency structure with Hermite prediction of high-frequency details. This provides a spectral view of “reuse versus forecast” caching that goes beyond purely temporal or spatial heuristics.

---

**3. Resolution of Main Technical Concerns**

**Generalization.** We showed that FreqCa is not tied to a specific backbone or step count. On a large U-Net, FreqCa outperforms DeepCache. On a video diffusion model, FreqCa matches or slightly outperforms prior accelerators while using less than 1% of their cache memory and avoiding out-of-memory issues. On both few-step distilled and 20-step baselines for Qwen-Image, FreqCa remains strong while TaylorSeer collapses.

**Spectral decomposition and CRF fidelity.** When we sweep the cutoff, we observe a broad stable range once it exceeds a small threshold, which supports a naturally bimodal spectrum and shows that FreqCa is insensitive to the exact cutoff choice. We use a 2D DCT in the main variant and FFT as an ablation. In FreqCa, CRF is treated as a compressed carrier on top of which we apply frequency decomposition and Hermite prediction. Experiments show high fidelity with O(1) cache and less than 1% of TaylorSeer’s memory usage.

**Predictor choice and acceleration limits.** We compared Hermite with other polynomial predictors and found that second-order Hermite provides the best trade-off between stability and quality at high acceleration. Intervals from 3 to 16 reveal a broad stable regime (approximately up to 6× speedup) with gradual degradation beyond this. We also clarified that on prediction steps the diffusion model is not executed. Instead, we reuse low-frequency features and predict high-frequency features via Hermite, with N and first\_enhance indicate which steps run the diffusion model and which are prediction only steps.

**Relation to prior work.** FreeU uses static frequency-related scaling for quality enhancement and does not perform explicit spectral analysis or reduce sampling cost. OmniCache focuses on when to cache (temporal scheduling), while BlockDance focuses on what to reuse (structured feature selection). In contrast, FreqCa focuses on what to cache and how to extrapolate, by reusing low-frequency information and predicting high-frequency information with O(1) CRF caching. This defines a frequency-aware axis that is orthogonal to these approaches and can be combined with them.

---

**4. Conclusion**

After rebuttal and discussion, all reviewers recognize FreqCa as a novel and impactful method that delivers significant acceleration and about 99% cache reduction across diverse diffusion models. The additional experiments and clarifications address concerns about generalization, robustness, compression fidelity, and predictor choice, and show that FreqCa remains effective in challenging regimes where competing methods degrade or run out of memory.

The main remaining concern is exposition, especially for itww, who explicitly states that the rating would be raised once the clarified explanation is incorporated. We are committed to these revisions and we hope you will consider an accept decision.

Best regards,

Authors of Paper #2446

---

### Note · Program_Chairs · 2026-01-17
**Submission Desk Rejected by Program Chairs**

The following references in this submission do not refer to real documents and/or have major errors in bibliographic information:

 Wen Wang, Qifeng Chen, Lvmin Zhang, Yue Ma, and Zexuan Yan. Gedit: A unified metric for evaluating instruction-based image editing. In Proceedings of the IEEE/CVF Conference on Computer Vision and Pattern Recognition, pp. 12345-12354, 2024.